# Beta Diffusion

**Mingyuan Zhou**,\* **Tianqi Chen, Zhendong Wang, and Huangjie Zheng**
The University of Texas at Austin
Austin, TX 78712

## Abstract

We introduce beta diffusion, a novel generative modeling method that integrates demasking and denoising to generate data within bounded ranges. Using scaled and shifted beta distributions, beta diffusion utilizes multiplicative transitions over time to create both forward and reverse diffusion processes, maintaining beta distributions in both the forward marginals and the reverse conditionals, given the data at any point in time. Unlike traditional diffusion-based generative models relying on additive Gaussian noise and reweighted evidence lower bounds (ELBOs), beta diffusion is multiplicative and optimized with KL-divergence upper bounds (KLUBs) derived from the convexity of the KL divergence. We demonstrate that the proposed KLUBs are more effective for optimizing beta diffusion compared to negative ELBOs, which can also be derived as the KLUBs of the same KL divergence with its two arguments swapped. The loss function of beta diffusion, expressed in terms of Bregman divergence, further supports the efficacy of KLUBs for optimization. Experimental results on both synthetic data and natural images demonstrate the unique capabilities of beta diffusion in generative modeling of range-bounded data and validate the effectiveness of KLUBs in optimizing diffusion models, thereby making them valuable additions to the family of diffusion-based generative models and the optimization techniques used to train them.

## 1 Introduction

Diffusion-based deep generative models have been gaining traction recently. One representative example is Gaussian diffusion [57, 59, 23, 60, 35] that uses a Gaussian Markov chain to gradually diffuse images into Gaussian noise for training. The learned reverse diffusion process, defined by a Gaussian Markov chain in reverse order, iteratively refines noisy inputs towards clean photo-realistic images. Gaussian diffusion can also be viewed from the lens of denoising score matching [28, 64, 59, 60] and stochastic differential equations [61]. They have shown remarkable success across a wide range of tasks, including but not limited to generating, restoring, and editing images [14, 24, 50, 53, 54, 70, 67], transforming 2D to 3D [49, 1], synthesizing audio [9, 37, 73], reinforcement learning [31, 66, 48], quantifying uncertainty [19], and designing drugs and proteins [56, 44, 32].

Constructing a diffusion-based generative model often follows a general recipe [57, 23, 35, 34]. The recipe involves three basic steps: First, defining a forward diffusion process that introduces noise into the data and corrupts it with decreasing signal-to-noise ratio (SNR) as time progresses from 0 to 1. Second, defining a reverse diffusion process that denoises the corrupted data as time reverses from 1 to 0. Third, discretizing the time interval from 0 to 1 into a finite number of intervals, and viewing the discretized forward and reverse processes as a fixed inference network and a learnable generator, respectively. Auto-encoding variational inference [36, 51] is then applied to optimize the parameters of the generator by minimizing a weighted negative ELBO that includes a Kullback–Leibler (KL) divergence-based loss term for each discretized reverse step.

---

\*Corresponding to: `mingyuan.zhou@mccombs.utexas.edu`
PyTorch code is available at: `https://github.com/mingyuanzhou/Beta-Diffusion`

37th Conference on Neural Information Processing Systems (NeurIPS 2023).

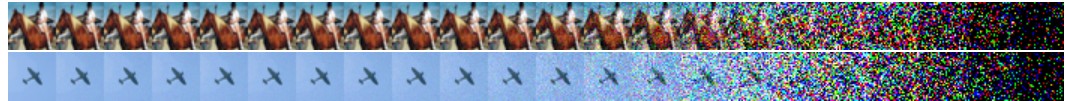

Figure 1: Illustration of the beta forward diffusion process for two example images. The first column displays the original images, while the other 21 columns display the images noised and masked by beta diffusion at time $t = 0, 0.05, \ldots, 1$, using $\eta = 10000$ and the sigmoid diffusion schedule with $c_0 = 10$ and $c_1 = -13$.

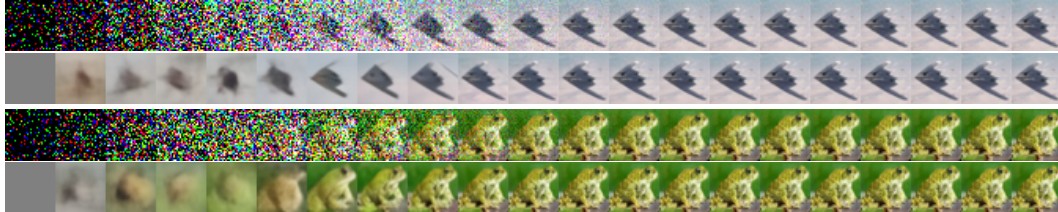

Figure 2: Illustration of reverse beta diffusion for two example generations. The time $t$ decreases from 1 to 0 when moving from left to right. In each image generation, the top row shows the trajectory of $z_t$ (rescaled for visualization), which has been demasked and denoised using reverse diffusion, whereas the bottom row shows $\hat{x}_0 = f_\theta(z_t, t)$, whose theoretical optimal solution is equal to $\mathbb{E}[x_0 \mid z_t]$. See Appendix D for more details.

Although the general diffusion-modeling recipe is simple in concept, it requires access to the corrupted data at any time during the forward diffusion process given a data observation, as well as the analytic form of the conditional posterior for any earlier time given both a data observation and its corrupted version at the present time. The latter requirement, according to Bayes' rule, implies access to the analytical form of the distribution of a corrupted data observation at the present time given its value at any previous time. Linear operations of the Gaussian distributions naturally satisfy these requirements since they are conjugate to themselves with respect to their mean parameters. This means that the marginal form and the conditional distribution of the mean remain Gaussian when two Gaussian distributions are mixed. Similarly, the requirements can be met under the categorical distribution [26, 2, 18, 27] and Poisson distribution [10]. However, few additional distributions are known to meet these requirements and it remains uncertain whether negative ELBO would be the preferred loss.

While previous works have primarily used Gaussian, categorical, or Poisson distribution-based diffusion processes, this paper introduces beta diffusion as a novel addition to the family of diffusion-based generative models. Beta diffusion is specifically designed to generate data within bounded ranges. Its forward diffusion process is defined by the application of beta distributions in a multiplicative manner, whereas its reverse diffusion process is characterized by the use of scaled and shifted beta distributions. Notably, the distribution at any point in time of the forward diffusion given a data observation remains a beta distribution. We illustrate the forward beta diffusion process in Figure 1, which simultaneously adds noise to and masks the data, and the reverse one in Figure 2, which iteratively performs demasking and denoising for data generation. We provide the details on how these images are obtained in Appendix D.

Since the KL divergence between two beta distributions is analytic, one can follow the general recipe to define a negative ELBO to optimize beta diffusion. However, our experiments show that minimizing the negative ELBO of beta diffusion can fail to optimally estimate the parameters of the reverse diffusion process. For each individual time-dependent KL loss term of the negative ELBO, examining it in terms of Bregman divergence [4] reveals that the model parameters and corrupted data are placed in its first and second arguments, respectively. However, to ensure that the optimal solution under the Bregman divergence agrees with the expectation of the clean data given the corrupted data, the order of the two arguments must be swapped.

By swapping the Bregman divergence's two arguments, we obtain an upper bound on the KL divergence from the joint distribution of corrupted observations in the reverse chain to that in the forward chain. This bound arises from the convexity of the KL divergence. In addition, there exists another Bregman divergence that upper bounds the KL divergence from the univariate marginal of a corrupted observation in the reverse chain to that in the forward chain. These two Bregman divergences, which can be derived either through KL divergence upper bounds (KLUBs) or the logarithmically convex beta function, share the same optimal solution but have distinct roles in targeting reverse accuracy at each step or counteracting accumulation errors over the course of reverse

diffusion. We further demonstrate that combining these two KLUBs presents a computationally viable substitution for a KLUB derived at the chain level, which upper bounds the KL divergence from the joint distribution of all latent variables in a forward diffusion chain to that of its reverse. Either KLUB works on its own, which is not unexpected as they both share the same optimal solutions in theory, but combining them could lead to the best overall performance. In beta diffusion, the KL divergence is asymmetric, which enables us to derive an alternative set of KLUBs by swapping its two arguments. We will demonstrate that these augment-swapped KLUBs, which will be referred to as AS-KLUBs, essentially reduce to negative ELBOs. In Gaussian diffusion, the KL divergence is often made symmetric, resulting in KLUBs that are equivalent to (weighted) negative ELBOs.

Our main contributions are the introduction of beta diffusion as a novel diffusion-based multiplicative generative model for range-bounded data, as well as the proposal of KLUBs as effective loss objectives for optimizing diffusion models, in place of (weighted) negative ELBOs. Additionally, we introduce the log-beta divergence, a Bregman divergence corresponding to the differentiable and strictly convex log-beta function, as a useful tool for analyzing KLUBs. These contributions enhance the existing family of diffusion-based generative models and provide a new perspective on optimizing them.

## 2 Beta Diffusion and Optimization via KLUBs

We begin by specifying the general requirements for constructing a diffusion-based generative model and establish the notation accordingly [57, 23, 35]. Let $x_0$ denote the observed data, and let $z_s$ and $z_t$ represent their corrupted versions at time $s$ and time $t$, respectively, where $0 < s < t < 1$. In the forward diffusion process, we require access to random samples from the marginal distribution $q(z_t \,|\, x_0)$ at any time $t$, as well as an analytical expression of the probability density function (PDF) of the conditional distribution $q(z_s \,|\, z_t, x_0)$ for any $s < t$.

The forward beta diffusion chain uses diffusion scheduling parameters $\alpha_t$ to control the decay of its expected value over the course of forward diffusion, given by $\mathbb{E}[z_t \,|\, x_0] = \alpha_t x_0$, and a positive concentration parameter $\eta$ to control the tightness of the diffusion process around its expected value. We typically set $\alpha_t$ to approach 1 and 0 as $t$ approaches 0 and 1, respectively, and satisfy the condition

$$1 \geq \alpha_0 > \alpha_s > \alpha_t > \alpha_1 \geq 0 \text{ for all } s \in (0, t), \, t \in (0, 1).$$

Let $\Gamma(\cdot)$ denote the gamma function and $B(\cdot, \cdot)$ denote the beta function. The beta distribution $\text{Beta}(x; a, b) = B(a, b)^{-1} x^{a-1} (1-x)^{b-1}$ is a member of the exponential family [6, 65, 7]. Its log partition function is a log-beta function as $\ln B(a, b) = \ln \Gamma(a) + \ln \Gamma(b) - \ln \Gamma(a + b)$, which is differentiable, and strictly convex on $(0, \infty)^2$ as a function of two variables [15]. As a result, the KL divergence between two beta distributions can be expressed as the Bregman divergence associated with the log-beta function. Specifically, as in Appendix A, one can show by their definitions that

$$\text{KL}(\text{Beta}(\alpha_p, \beta_p) || \text{Beta}(\alpha_q, \beta_q)) = \ln \frac{B(\alpha_q, \beta_q)}{B(\alpha_p, \beta_p)} - (\alpha_q - \alpha_p, \beta_q - \beta_p) \begin{pmatrix} \nabla_\alpha \ln B(\alpha_p, \beta_p) \\ \nabla_\beta \ln B(\alpha_p, \beta_p) \end{pmatrix}$$

$$= D_{\ln B(a,b)}((\alpha_q, \beta_q), (\alpha_p, \beta_p)). \tag{1}$$

We refer to the above Bregman divergence as the log-beta divergence. Moreover, if $(\alpha_q, \beta_q)$ are random variables, applying Proposition 1 of Banerjee et al. [4], we can conclude that the optimal value of $(\alpha_p, \beta_p)$ that minimizes this log-beta divergence is $(\alpha_p^*, \beta_p^*) = \mathbb{E}[(\alpha_q, \beta_q)]$.

Next, we introduce a conditional bivariate beta distribution, which given a data observation has (scaled and shifted) beta distributions for not only its two marginals but also two conditionals. These properties are important for developing the proposed diffusion model with multiplicative transitions.

### 2.1 Conditional Bivariate Beta Distribution

We first present the conditional bivariate beta distribution in the following Lemma, which generalizes previous results on the distribution of the product of independent beta random variables [30, 38, 33].

**Lemma 1** (Conditional Beta Bivariate Distribution). *Denote $(z_s, z_t)$ as variables over a pair of time points $(s, t)$, with $0 < s < t < 1$. Given a random sample $x_0 \in (0, 1)$ from a probability-valued data distribution $p_{data}(x_0)$, we define a conditional bivariate beta distribution over $(z_s, z_t)$ with PDF:*

$$q(z_s, z_t \,|\, x_0) = \frac{\Gamma(\eta)}{\Gamma(\eta \alpha_t x_0) \Gamma(\eta(1 - \alpha_s x_0)) \Gamma(\eta(\alpha_s - \alpha_t) x_0)} \frac{z_t^{\eta \alpha_t x_0 - 1} (1 - z_s)^{\eta(1 - \alpha_s x_0) - 1}}{(z_s - z_t)^{1 - \eta(\alpha_s - \alpha_t) x_0}}. \tag{2}$$

*Marginals: Given $x_0$, the two univariate marginals of this distribution are both beta distributed as*

$$q(z_s \,|\, x_0) = \text{Beta}(\eta\alpha_s x_0, \eta(1 - \alpha_s x_0)), \tag{3}$$

$$q(z_t \,|\, x_0) = \text{Beta}(\eta\alpha_t x_0, \eta(1 - \alpha_t x_0)). \tag{4}$$

*Conditionals: Given $x_0$, a random sample $(z_t, z_s)$ from this distribution can be either generated in forward order, by multiplying a beta variable from (3) with beta variable $\pi_{s\to t}$, as*

$$z_t = z_s \pi_{s\to t}, \quad \pi_{s\to t} \sim \text{Beta}(\eta\alpha_t x_0, \eta(\alpha_s - \alpha_t)x_0), \quad z_s \sim q(z_s \,|\, x_0), \tag{5}$$

*or generated in reverse order, by combining a beta variable from (4) with beta variable $p_{s\gets t}$, as*

$$z_s = z_t + (1 - z_t)p_{s\gets t}, \quad p_{s\gets t} \sim \text{Beta}(\eta(\alpha_s - \alpha_t)x_0, \eta(1 - \alpha_s x_0)), \quad z_t \sim q(z_t \,|\, x_0). \tag{6}$$

The proof starts with applying change of variables to obtain two scaled and shifted beta distributions

$$q(z_t \,|\, z_s, x_0) = \frac{1}{z_s}\text{Beta}\left(\frac{z_t}{z_s}; \eta\alpha_t x_0, \eta(\alpha_s - \alpha_t)x_0\right), \tag{7}$$

$$q(z_s \,|\, z_t, x_0) = \frac{1}{1-z_t}\text{Beta}\left(\frac{z_s-z_t}{1-z_t}; \eta(\alpha_s - \alpha_t)x_0, \eta(1 - \alpha_s x_0)\right), \tag{8}$$

and then takes the products of them with their corresponding marginals, given by (3) and (4), respectively, to show that the PDF of the joint distribution defined in (5) and that defined in (6) are both equal to the PDF of $q(z_s, z_t \,|\, x_0)$ defined in (2). The detailed proof is provided in Appendix B. To ensure numerical accuracy, we will calculate $z_s = z_t + (1 - z_t)p_{s\gets t}$ in (6) in the logit space as

$$\text{logit}(z_s) = \ln\left(e^{\text{logit}(z_t)} + e^{\text{logit}(p_{s\gets t})} + e^{\text{logit}(z_t)+\text{logit}(p_{s\gets t})}\right). \tag{9}$$

## 2.2 Continuous Beta Diffusion

**Forward Beta Diffusion.** We can use the conditional bivariate beta distribution to construct a forward beta diffusion chain, beginning with the beta distribution from (3) and proceeding with the scaled beta distribution from (7). The marginal at any time $t$ for a given data observation $x_0$, as shown in (4), stays as beta-distributed in the forward chain. For the beta distribution given by (4), we have

$$\mathbb{E}[z_t \,|\, x_0] = \alpha_t x_0, \quad \text{var}[z_t \,|\, x_0] = \frac{(\alpha_t x_0)(1-\alpha_t x_0)}{\eta+1}, \quad \text{SNR}_t = \left(\frac{\mathbb{E}[z_t \,|\, x_0]}{\text{std}[z_t \,|\, x_0]}\right)^2 = \frac{\alpha_t x_0(\eta+1)}{1-\alpha_t x_0}.$$

Thus when $\alpha_t$ approaches 0 (*i.e.*, $t \to 1$), both $z_t$ and $\text{SNR}_t$ are shrunk towards 0, and if $\alpha_1 = 0$, we have $z_1 \sim \text{Beta}(0, \eta)$, a degenerate beta distribution that becomes a unit point mass at 0.

**Infinite Divisibility.** We consider beta diffusion as a form of continuous diffusion, as its forward chain is infinitely divisible given $x_0$. This means that for any time $k \in (s, t)$, we can perform forward diffusion from $z_s$ to $z_t$ by first setting $z_k = z_s \pi_{s\to k}$, where $\pi_{s\to k} \sim \text{Beta}(\eta\alpha_k x_0, \eta(\alpha_k - \alpha_s)x_0)$, and then setting $z_t = z_k \pi_{k\to t}$, where $\pi_{k\to t} \sim \text{Beta}(\eta\alpha_t x_0, \eta(\alpha_t - \alpha_k)x_0)$. The same approach can be used to show the infinite divisibility of reverse beta diffusion given $x_0$.

**Reverse Beta Diffusion.** We follow Gaussian diffusion to use $q(z_s \,|\, z_t, x_0)$ to help define $p_\theta(z_s \,|\, z_t)$ [58, 35, 72]. To construct a reverse beta diffusion chain, we will first need to learn how to reverse from $z_t$ to $z_s$, where $s < t$. If we know the $x_0$ used to sample $z_t$ as in (3), then we can readily apply the conditional in (8) to sample $z_s$. Since this information is unavailable during inference, we make a weaker assumption that we can exactly sample from $z_t \sim q(z_t) = \mathbb{E}_{x_0\sim p_{data}}[q(z_t \,|\, x_0)]$, which is the "$x_0$-free" univariate marginal at time $t$. It is straightforward to sample during training but must be approximated during inference. Under this weaker assumption on $q(z_t)$, utilizing (8) but replacing its true $x_0$ with an approximation $\hat{x}_0 = f_\theta(z_t, t)$, where $f_\theta$ denotes the learned generator parameterized by $\theta$, we introduce our "$x_0$-free" and hence "causal" time-reversal distribution as

$$p_\theta(z_s \,|\, z_t) = q(z_s \,|\, z_t, \hat{x}_0 = f_\theta(z_t, t)). \tag{10}$$

## 2.3 Optimization via KLUBs and Log-Beta Divergence

**KLUB for Time Reversal.** The time-reversal distribution $p_\theta(z_s \,|\, z_t)$ reaches its optimal when its product with $q(z_t)$ becomes equivalent to $q(z_s, z_t) = \mathbb{E}_{x_0\sim p_{data}}[q(z_s, z_t \,|\, x_0)]$, which is a marginal bivariate distribution that is "$x_0$-free." Thus we propose to optimize $\theta$ by minimizing $\text{KL}(p_\theta(z_s \,|\, z_t)q(z_t)||q(z_s, z_t))$ in theory but introduce a surrogate loss in practice:

**Lemma 2** (KLUB (conditional))**.** *The KL divergence from $q(z_s, z_t)$ to $p_\theta(z_s \mid z_t)q(z_t)$, two "$x_0$-free" joint distributions defined by forward and reverse diffusions, respectively, can be upper bounded:*

$$\mathrm{KL}(p_\theta(z_s \mid z_t)q(z_t)||q(z_s, z_t)) \le \mathrm{KLUB}_{s,t} = \mathbb{E}_{(z_t, x_0) \sim q(z_t \mid x_0)p_{data}(x_0)}[\mathrm{KLUB}(s, z_t, x_0)], \quad (11)$$

$$\mathrm{KLUB}(s, z_t, x_0) = \mathrm{KL}(q(z_s \mid z_t, \hat{x}_0 = f_\theta(z_t, t))||q(z_s \mid z_t, x_0)). \quad (12)$$

The proof in Appendix B utilizes the equation $p_\theta(z_s \mid z_t)q(z_t) = \mathbb{E}_{x_0 \sim p_{data}}[p_\theta(z_s \mid z_t)q(z_t \mid x_0)]$ and then applies the convexity of the KL divergence [11, 74] and the definition in (10).

**Log-Beta Divergence.** To find out the optimal solution under KLUB, following (1), we can express $\mathrm{KLUB}(s, z_t, x_0)$ given by (12) as a log-beta divergence as

$$D_{\ln B(a,b)}\big\{ [\eta(\alpha_s - \alpha_t)x_0, \eta(1 - \alpha_s x_0)], \ [\eta(\alpha_s - \alpha_t)f_\theta(z_t, t), \eta(1 - \alpha_s f_\theta(z_t, t))] \big\}. \quad (13)$$

We note $\mathrm{KLUB}_{s,t}$ defined in (11) can also be written as $\mathbb{E}_{z_t \sim q(z_t)}\mathbb{E}_{x_0 \sim q(x_0 \mid z_t)}[\mathrm{KLUB}(s, z_t, x_0)]$, where the log-beta divergence for $\mathrm{KLUB}(s, z_t, x_0)$, defined as in (13), includes $x_0 \sim q(x_0 \mid z_t)$ in its first argument and the generator $f_\theta(z_t, t)$ in its second argument. Therefore, applying Proposition 1 of Banerjee et al. [4], we have the following Lemma.

**Lemma 3.** *The objective $\mathrm{KLUB}_{s,t}$ defined in* (11) *for any $s < t$ is minimized when*

$$f_{\theta^*}(z_t, t) = \mathbb{E}[x_0 \mid z_t] = \mathbb{E}_{x_0 \sim q(x_0 \mid z_t)}[x_0] \quad \text{for all } z_t \sim q(z_t). \quad (14)$$

Thus under the $\mathrm{KLUB}_{s,t}$-optimized $\theta^*$, we have $p_{\theta^*}(z_s \mid z_t) = q(z_s \mid z_t, \mathbb{E}[x_0 \mid z_t])$, which is different from the optimal solution of the original KL loss in (11), which is $p_\theta^*(z_s \mid z_t) = q(z_s, z_t)/q(z_t) = q(z_s \mid z_t) = \mathbb{E}_{x_0 \sim q(x_0 \mid z_t)}[q(z_s \mid z_t, x_0)]$. It is interesting to note that they only differ on whether the expectation is carried out inside or outside the conditional posterior.

In practice, we need to control the gap between $p_{\theta^*}(z_s \mid z_t)$ and $q(z_s \mid z_t)$ and hence $s$ needs to be close to $t$. Furthermore, the assumption of having access to unbiased samples from the true marginal $q(z_t)$ is also rarely met. Thus we need to discretize the time from 1 to $t$ into sufficiently fine intervals and perform time-reversal sampling over these intervals. Specifically, we can start with $z_1 = 0$ and iterate (10) over these intervals to obtain an approximate sample from $z_t \sim q(z_t)$. However, the error could accumulate along the way from $z_1$ to $z_t$, to which we present a solution below.

**KLUB for Error Accumulation Control.** To counteract error accumulation during time reversal, we propose to approximate the true marginal $q(z_t')$ using a "distribution-cycle-consistency" approach. This involves feeding a random sample $z_t$ from $q(z_t)$ into the generator $f_\theta$, followed by the forward marginal $q(z_t' \mid \hat{x}_0 = f_\theta(z_t, t))$, with the aim of recovering the distribution $q(z_t')$ itself. Specifically, we propose to approximate $q(z_t')$ with $p_\theta(z_t') := \mathbb{E}_{z_t \sim q(z_t)}[q(z_t' \mid \hat{x}_0 = f_\theta(z_t, t))]$ by minimizing $\mathrm{KL}(p_\theta(z_t')||q(z_t'))$ in theory, but introducing a surrogate loss in practice:

**Lemma 4** (KLUB (marginal))**.** *The KL divergence $\mathrm{KL}(p_\theta(z_t')||q(z_t'))$ can be upper bounded:*

$$\mathrm{KL}(p_\theta(z_t')||q(z_t')) \le \mathrm{KLUB}_t = \mathbb{E}_{(z_t, x_0) \sim q(z_t \mid x_0)p_{data}(x_0)}[\mathrm{KLUB}(z_t, x_0)], \quad (15)$$

$$\mathrm{KLUB}(z_t, x_0) = \mathrm{KL}(q(z_t' \mid f_\theta(z_t, t))||q(z_t' \mid x_0)). \quad (16)$$

The proof in Appendix B utilizes the fact that $q(z_t') = \mathbb{E}_{(z_t, x_0) \sim q(z_t \mid x_0)p_{data}(x_0)}[q(z_t' \mid x_0)]$.

Note that the mathematical definition of KLUB is reused throughout the paper and can refer to any of the equations (11), (12), (15), or (16) depending on the context. Similar to previous analysis, we have

$$\mathrm{KLUB}(z_t, x_0) = D_{\ln B(a,b)}\big\{ [\eta\alpha_t x_0, \eta(1 - \alpha_t x_0)], \ [\eta\alpha_t f_\theta(z_t, t), \eta(1 - \alpha_t f_\theta(z_t, t))] \big\}, \quad (17)$$

and can conclude with the following Lemma.

**Lemma 5.** $\mathrm{KLUB}_t$ *in* (15) *is optimized when the same optimal solution given by* (14) *is met.*

**Optimization via KLUBs.** With the two KLUBs for both time reversal and error accumulation control, whose optimal solutions are the same as in (14), we are ready to optimize the generator $f_\theta$ via stochastic gradient descent (SGD). Specifically, denoting $\omega, \pi \in [0, 1]$ as two weight coefficients, the loss term for the $i$th data observation $x_0^{(i)}$ in a mini-batch can be computed as

$$\mathcal{L}_i = \omega\mathrm{KLUB}(s_i, z_{t_i}, x_0^{(i)}) + (1 - \omega)\mathrm{KLUB}(z_{t_i}, x_0^{(i)}), \quad (18)$$

$$z_{t_i} \sim q(z_{t_i} \mid x_0^{(i)}) = \mathrm{Beta}(\eta\alpha_{t_i} x_0^{(i)}, \eta(1 - \alpha_{t_i} x_0^{(i)})), \ s_i = \pi t_i, \quad t_i \sim \mathrm{Unif}(0, 1).$$

## 2.4 Discretized Beta Diffusion for Generation of Range-Bounded Data

For generating range-bounded data, we discretize the beta diffusion chain. Denote $z_{t_0} = 1$ and let $t_j$ increase with $j$. Repeating (5) for $T$ times, we define a discretized forward beta diffusion chain:

$$q(z_{t_{1:T}} \,|\, x_0) = \prod_{j=1}^{T} q(z_{t_j} \,|\, z_{t_{j-1}}, x_0) = \prod_{j=1}^{T} \frac{1}{z_{t_{j-1}}} \text{Beta}\left(\frac{z_{t_j}}{z_{t_{j-1}}}; \eta \alpha_{t_j} x_0, \eta(\alpha_{t_{j-1}} - \alpha_{t_j}) x_0\right). \quad (19)$$

A notable feature of (19) is that the marginal at any discrete time step $t_j$ follows a beta distribution, similarly defined as in (4). We also note while $q(z_{t_{1:T}} \,|\, x_0)$ defines a Markov chain, the marginal

$$q(z_{t_{1:T}}) = \mathbb{E}_{x_0 \sim p_{data}(x_0)}[q(z_{t_{1:T}} \,|\, x_0)] \quad (20)$$

in general does not. Unlike in beta diffusion, where the transitions between $z_t$ and $z_{t-1}$ are applied multiplicatively, in Gaussian diffusion, the transitions between $z_t$ and $z_{t-1}$ are related to each other additively and $z_{t_{1:T}}$ forms a Markov chain regardless of whether $x_0$ is marginalized out.

The discretized forward beta diffusion chain given by (19) is reversible assuming knowing $x_0$. This means given $x_0$, it can be equivalently sampled in reverse order by first sampling $z_{t_T} \sim q(z_{t_T} \,|\, x_0) = \text{Beta}(\eta \alpha_{t_T} x_0, \eta(1 - \alpha_{t_T} x_0))$ and then repeating (8) for $t_T, \ldots, t_2$, with PDF $q(z_{t_{1:T}} \,|\, x_0) = q(z_{t_T} \,|\, x_0) \prod_{j=2}^{T} \frac{1}{1 - z_{t_j}} \text{Beta}\left(\frac{z_{t_{j-1}} - z_{t_j}}{1 - z_{t_j}}; \eta(\alpha_{t_{j-1}} - \alpha_{t_j}) x_0, \eta(1 - \alpha_{t_{j-1}} x_0)\right)$. This non-causal chain, while not useful by itself, serves as a blueprint for approximate generation.

Specifically, we approximate the marginal given by (20) with a Markov chain in reverse order as

$$p_\theta(z_{t_{1:T}}) = p_{prior}(z_{t_T}) \prod_{j=2}^{T} p_\theta(z_{t_{j-1}} \,|\, z_{t_j}) = p_{prior}(z_{t_T}) \prod_{j=2}^{T} q(z_{t_{j-1}} \,|\, z_{t_j}, f_\theta(z_{t_j}, \alpha_{t_j})). \quad (21)$$

To start the reverse process, we choose to approximate $q(z_{t_T}) = \mathbb{E}_{x_0 \sim p_{data}(x_0)}[q(z_{t_T} \,|\, x_0)]$ with $p_{prior}(z_{t_T}) = q(z_{t_T} \,|\, \mathbb{E}[x_0])$, which means we let $z_{t_T} \sim \text{Beta}(\eta \alpha_{t_T} \mathbb{E}[x_0], \eta(1 - \alpha_{t_T} \mathbb{E}[x_0]))$. To sample $z_{t_{1:T-1}}$, we use the remaining terms in (21), which are scaled and shifted beta distributions that are specified as in (8) and can be sampled as in (6) and (9).

**KLUB for Discretized Beta Diffusion.** An optimized generator is expected to make the "$x_0$-free" joint distribution over all $T$ steps of the discretized reverse beta diffusion chain, expressed as $p_\theta(z_{t_{1:T}})$, to approach that of the discretized forward chain, expressed as $q(z_{t_{1:T}})$. Thus an optimized $\theta$ is desired to minimize the KL divergence $\text{KL}(p_\theta(z_{t_{1:T}}) || q(z_{t_{1:T}}))$. While this KL loss is in general intractable to compute, it can also be bounded using the KLUB shown as follows.

**Lemma 6** (KLUB for discretized diffusion chain). $\text{KL}(p_\theta(z_{t_{1:T}}) || q(z_{t_{1:T}}))$ *is upper bounded by*

$$\text{KLUB} = \mathbb{E}_{x_0 \sim p_{data}(x_0)} \left[\text{KL}(p_{prior}(z_{t_T}) || q(z_{t_T} \,|\, x_0))\right] + \sum_{j=2}^{T} \widetilde{\text{KLUB}}_{t_{j-1}, t_j} \quad where \quad (22)$$

$$\widetilde{\text{KLUB}}_{s,t} = \mathbb{E}_{(z_t, x_0) \sim p_\theta(z_t \,|\, x_0) p_{data}(x_0)}[\text{KLUB}(s, z_t, x_0)], \quad \text{KLUB}(s, z_t, x_0) = \text{KL}(p_\theta(z_s \,|\, z_t) || q(z_s \,|\, z_t, x_0))].$$

We provide the proof in Appendix B. We note Lemma 6 is a general statement applicable for any diffusion models with a discrete forward chain $q(z_{t_{1:T}} \,|\, x_0) = \prod_{j=1}^{T} q(z_{t_j} \,|\, z_{t_{j-1}}, x_0)$ and a discrete reverse chain $p_\theta(z_{t_{1:T}}) = p_{prior}(z_{t_T}) \prod_{j=2}^{T} p_\theta(z_{t_{j-1}} \,|\, z_{t_j})$. To estimate the KLUB in (22), however, during training, one would need to sample $z_{t_j} \sim p_\theta(z_{t_j} \,|\, x_0) \propto p(x_0 \,|\, z_{t_j}) p_\theta(z_{t_j})$, which is often infeasible. If we replace $z_{t_j} \sim p_\theta(z_{t_j} \,|\, x_0)$ with $z_{t_j} \sim q(z_{t_j} \,|\, x_0)$, then $\widetilde{\text{KLUB}}_{s,t}$ becomes the same as $\text{KLUB}_{s,t}$ given by (11), and $\text{KLUB}_t$ given by (15) can be considered to remedy the impact of approximating $p_\theta(z_{t_j})$ with $q(z_{t_j})$. Therefore, we can consider the combination of $\text{KLUB}_{s,t}$ given by (11) and $\text{KLUB}_t$ given by (15) as a computationally viable solution to compute $\widetilde{\text{KLUB}}_{s,t}$, which hence justifies the use of the loss in (18) to optimize the discretized reverse beta diffusion chain. We summarize the training and sampling algorithms of beta diffusion in Algorithms 1 and 2, respectively.

## 2.5 Argument-swapped KLUBs and Negative ELBOs

We note that in theory, instead of the KL divergences in (11) and (15), $f_\theta$ can also be optimized using two argument-swapped KL divergences: $\text{KL}(q(z_t, z_s) || p_\theta(z_s | z_t) q(z_t))$ and $\text{KL}(q(z_t') || p_\theta(z_t'))$. By the same analysis, these KL divergences can also be bounded by KLUBs and log-beta divergences that are equivalent to the previous ones, but with swapped arguments. The argument-swapped KLUBs and log-beta divergences will be shown to be closely related to optimizing a discretized beta reverse

diffusion chain via −ELBO, but they do not guarantee an optimal solution that satisfies (14) in beta diffusion and are found to provide clearly inferior empirical performance.

**Negative ELBO for Discretized Beta Diffusion.** As an alternative to KLUB, one can also consider following the convention in diffusion modeling to minimize the negative ELBO, expressed as

$$- \mathbb{E}_{x_0 \sim p_{data}(x_0)} \ln p_\theta(x_0) \leq -\text{ELBO} = \mathbb{E}_{x_0 \sim p_{data}(x_0)} \mathbb{E}_{q(z_{t_{1:T}} \mid x_0)} \left[ -\ln \frac{p(x_0 \mid z_{t_{1:T}}) p_\theta(z_{t_{1:T}})}{q(z_{t_{1:T}} \mid x_0)} \right]$$

$$= -\mathbb{E}_{x_0} \mathbb{E}_{q(z_{t_1} \mid x_0)} \ln p(x_0 \mid z_{t_1}) + \mathbb{E}_{x_0} \text{KL}[q(z_{t_T} \mid x_0) || p_{prior}(z_{t_T})] + \sum_{j=2}^{T} \mathbb{E}_{x_0} \mathbb{E}_{q(z_{t_j} \mid x_0)} [L(t_{j-1}, z_{t_j}, x_0)],$$

where the first two terms are often ignored and the focus is placed on the remaining $T - 2$ terms as

$$L(t_{j-1}, z_{t_j}, x_0) = \text{KL}(q(z_{t_{j-1}} \mid z_{t_j}, x_0) || q(z_{t_{j-1}} \mid z_{t_j}, \hat{x}_0 = f_\theta(z_{t_j}, t_j)))$$

$$= D_{\ln B(a,b)} \Big\{ \big[ \eta(\alpha_{t_{j-1}} - \alpha_{t_j}) f_\theta(z_{t_j}, t_j), \eta(1 - \alpha_{t_{j-1}} f_\theta(z_{t_j}, t_j)) \big], \ \big[ \eta(\alpha_{t_{j-1}} - \alpha_{t_j}) x_0, \eta(1 - \alpha_{t_{j-1}} x_0) \big] \Big\}. \quad (23)$$

**Lemma 7** (−ELBO and argument-swapped KLUB). *Optimizing the generator $f_\theta$ with −ELBO is equivalent to using an upper-bound for the augment-swapped KL divergence $\text{KL}(q(z_{t_{1:T}}) || p_\theta(z_{t_{1:T}}))$.*

The proof in Appendix B relies on the convex nature of both the KL divergence and the negative logarithmic function. We find for beta diffusion, optimizing $\text{KL}(p_\theta(z_{t_{1:T}}) || q(z_{t_{1:T}}))$ via the proposed KLUBs is clearly preferred to optimizing $\text{KL}(q(z_{t_{1:T}}) || p_\theta(z_{t_{1:T}}))$ via −ELBOs (*i.e.*, augment-swapped KLUBs) and leads to stable and satisfactory performance.

**KLUBs and (Weighted) Negative ELBOs for Gaussian Diffusion.** We note KLUBs are directly applicable to Gaussian diffusion, but they may not result in new optimization algorithms for Gaussian diffusion that drastically differ from the weighted ELBO, which weighs the KL terms using the corresponding SNRs. Moreover, whether the default or argument-swapped KLUBs are used typically does not make any difference in Gaussian diffusion and would result in the same squared error-based Bregman divergence. We provide the derivation of the (weighted) ELBOs from the lens of KLUBs in Appendix C, providing theoretical support for Gaussian diffusion to use the SNR weighted ELBO [23, 35, 20], which was often considered as a heuristic but crucial modification of ELBO.

## 3 Related Work, Limitations, and Future Directions

Various diffusion processes, including Gaussian, categorical, Poisson, and beta diffusions, employ specific distributions in both forward and reverse sampling. Gaussian diffusion starts at $\mathcal{N}(0, 1)$ in its reverse process, while both Poisson and beta diffusion start at 0. Beta diffusion's reverse sampling is a monotonically non-decreasing process, similar to Poisson diffusion, but while Poisson diffusion takes count-valued discrete jumps, beta diffusion takes probability-valued continuous jumps. A future direction involves extending beta diffusion to encompass the exponential family [6, 65, 45, 7].

Several recent works have explored alternative diffusion processes closely related to Gaussian diffusion. Cold diffusion by Bansal et al. [5] builds models around arbitrary image transformations instead of Gaussian corruption, but it still relies on $L_1$ loss, resembling Gaussian diffusion's squared Euclidean distance. Rissanen et al. [52] propose an inverse heat dispersion-based diffusion process that reverses the heat equation using inductive biases in Gaussian diffusion-like models. Soft diffusion by Daras et al. [12] uses linear corruption processes like Gaussian blur and masking. Blurring diffusion by Hoogeboom and Salimans [25] shows that blurring (or heat dissipation) can be equivalently defined using a Gaussian diffusion process with non-isotropic noise and proposes to incorporate blurring into Gaussian diffusion. These alternative diffusion processes share similarities with Gaussian diffusion in loss definition and the use of Gaussian-based reverse diffusion for generation. By contrast, beta diffusion is distinct from all of them in forward diffusion, training loss, and reverse diffusion.

A concurrent work by Avdeyev et al. [3] utilizes the Jacobi diffusion process for discrete data diffusion models. Unlike Gaussian diffusion's SDE definition, the Jacobi diffusion process in Avdeyev et al. [3] is defined by the SDE $dx = \frac{s}{2}[a(1 - x) - bx]dt + \sqrt{sx(1 - x)}dw$, with $x \in [0, 1]$ and $s, a, b > 0$. The stationary distribution is a univariate beta distribution $\text{Beta}(a, b)$. Beta diffusion and the Jacobi process are related to the beta distribution, but they differ in several aspects: Beta diffusion ends its forward process at $\text{Beta}(0, \eta)$, a unit point mass at 0, not a $\text{Beta}(a, b)$ random variable. The marginal distribution of beta diffusion at time $t$ is expressed as $q(z_t \mid x_0) \sim \text{Beta}(\eta \alpha_t x_0, \eta(1 - \alpha_t x_0))$, while the Jacobi diffusion process involves an infinite sum. Potential connections between beta diffusion and the Jacobi process under specific parameterizations are worth further investigation.

Several recent studies have been actively exploring the adaptation of diffusion models to constrained scenarios [13, 40, 41, 17], where data is bounded within specific ranges or constrained to particular manifolds. These approaches are all rooted in the framework of Gaussian diffusion, which involves the incorporation of additive Gaussian noise. In sharp contrast, beta diffusion introduces a distinct perspective by incorporating multiplicative noise, leading to the emergence of an inherently hypercubic-constrained diffusion model that offers new development and exploration opportunities.

Classifier-free guidance (CFG), often used in conjunction with heuristic clipping, is a widely used technique to perform conditional generation with Gaussian diffusion [22, 41]. Beta diffusion offers the potential for a seamless integration of CFG by directly applying it to the logit space, the operating space of its $f_\theta$ network, thereby eliminating the necessity for heuristic clipping.

To adapt Gaussian diffusion for high-resolution images, a common approach is to perform diffusion within the latent space of an auto-encoder [53, 62]. A promising avenue to explore is the incorporation of sigmoid or tanh activation functions in the encoder's final layer. This modification would establish a bounded latent space conducive to applying beta diffusion, ultimately leading to the development of latent beta diffusion tailored for high-resolution image generation.

One limitation of beta diffusion is that its training is computationally expensive and data-intensive, akin to Gaussian diffusion. Specifically, with four Nvidia RTX A5000 GPUs, beta diffusion and Gaussian diffusion (VP-EDM) both take approximately 1.46 seconds to process 1000 images of size $32 \times 32 \times 3$. Processing 200 million CIFAR-10 images, the default number required to reproduce the results of VP-EDM, would thus take over 80 hours. Several recent works have explored different techniques to make Gaussian diffusion faster and/or more data efficient in training [68, 20, 76, 71]. It is worth exploring how to adapt these methods to enhance the training efficiency of beta diffusion.

Beta diffusion has comparable sampling costs to Gaussian diffusion with the same NFE. However, various methods have been developed to accelerate the generation of Gaussian diffusion, including combining it with VAEs, GANs, or conditional transport [77] for faster generation [72, 47, 78, 69], distilling the reverse diffusion chains [43, 55, 76], utilizing reinforcement learning [16], and transforming the SDE associated with Gaussian diffusion into an ODE, followed by fast ODE solvers [58, 39, 42, 75, 34]. Given these existing acceleration techniques for Gaussian diffusion, it is worth exploring their generalization to enhance the sampling efficiency of beta diffusion.

Beta diffusion raises concerns regarding potential negative societal impact when trained on image datasets curated with ill intentions. This issue is not exclusive to beta diffusion but applies to diffusion-based generative models as a whole. It is crucial to address how we can leverage these models for the betterment of society while mitigating any potential negative consequences.

# 4 Experiments

The training and sampling algorithms for beta diffusion are described in detail in Algorithms 1 and 2, respectively, in the Appendix. Our experiments, conducted on two synthetic data and the CIFAR10 images, primarily aim to showcase beta diffusion's effectiveness in generating range-bounded data. We also underscore the superiority of KLUBs over negative ELBOs as effective optimization objectives for optimizing beta diffusion. Additionally, we highlight the differences between beta and Gaussian diffusion, specifically in whether the data are generated through additive or multiplicative transforms and their ability to model the mixture of range-bounded distributions with disjoint supports.

We compare the performance of "Gauss ELBO," "Beta ELBO," and "Beta KLUB," which respectively correspond to a Gaussian diffusion model optimized with the SNR weighted negative ELBO [23, 35], a beta diffusion model optimized with the proposed KLUB loss defined in (18) but with the two arguments inside each KL term swapped, and a beta diffusion model optimized with the proposed KLUB loss defined in (18). On CIFAR-10, we also evaluate beta diffusion alongside a range of non-Gaussian or Gaussian-like diffusion models for comparison.

## 4.1 Synthetic Data

We consider a discrete distribution that consists of an equal mixture of five unit point masses located at $x_0 \in \mathcal{D} = \{1/7, 2/7, 3/7, 4/7, 5/7\}$. We would like to highlight that a unit point mass can also be seen as an extreme case of range-bounded data, where the range is zero. Despite being simple,

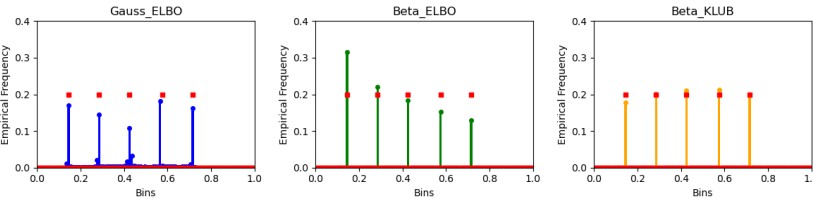

Figure 3: Comparison of the true PMF, marked in red square ■, and the empirical PMFs of three different methods—Gauss ELBO, Beta ELBO, and Beta KLUB— calculated over 100 equal-sized bins between 0 and 1. Each empirical PMF is marked in solid dot ●.

this data could be challenging to model by a continuous distribution, as it would require the generator to concentrate its continuous-valued generations on these five discrete points.

We follow previous works to choose the beta linear diffusion schedule as $\alpha_t = e^{-\frac{1}{2}\beta_d t^2 - \beta_{\min} t}$, where $\beta_d = 19.9$ and $\beta_{\min} = 0.1$. This schedule, widely used by Gaussian diffusion [23, 61, 34], is applied consistently across all experiments conducted on synthetic data. We set $\eta = 10000$, $\pi = 0.95$, and $\omega = 0.5$. As the data already falls within the range of 0 to 1, necessitating neither scaling nor shifting, we set $S_{cale} = 1$ and $S_{hift} = 0$. We use the same structured generator $f_\theta$ for both Gaussian and beta diffusion. We choose 20-dimensional sinusoidal position embeddings [63], with the positions set as $1000t$. The network is an MLP structured as (21-256)-ReLU-(256-256)-ReLU-(256-1). We utilize the Adam optimizer with a learning rate of 5e-4 and a mini-batch size of 1000.

For data generation, we set NFE = 200. We provide the generation results in Figure 3, which shows the true probability mass function (PMF) of the discrete distribution and the empirical PMFs over 100 equal-sized bins between 0 and 1. Each empirical PMF is computed based on 100k random data points generated by the model trained after 400k iterations. It is clear from Figure 3 that "Gauss ELBO" is the worst in terms of mis-aligning the data supports and placing its data into zero-density regions; "Beta ELBO" is the worst in terms of systematically overestimating the density at smaller-valued supports; whereas "Beta KLUB" reaches the best compromise between accurately identifying the data supports and maintaining correct density ratios between different supports.

In Appendix E, we further provide quantitative performance comparisons between different diffusion models and conduct an ablation study between KLUB and its two variants for beta diffusion: "KLUB Conditional" and "KLUB Marginal," corresponding to $\omega = 1$ and $\omega = 0$, respectively, in the loss given by (18). Additionally, we evaluate beta diffusion on another synthetic data, which comes from a mixture of range-bounded continuous distributions and point masses supported on disjoint regions.

## 4.2 Experiments on Image Generation

We employ the CIFAR-10 dataset and build upon VP-EDM [34] as the foundation of our codebase. Our initial foray into applying beta diffusion to generative modeling of natural images closely mirrors the settings of Gaussian diffusion, including the choice of the generator's network architecture. We introduce a sigmoid diffusion schedule defined as $\alpha_t = 1/(1 + e^{-c_0 - (c_1 - c_0)t})$, which has been observed to offer greater flexibility than the beta linear schedule for image generation. This schedule bears resemblance to the sigmoid-based one introduced for Gaussian diffusion [35, 29]. We configure the parameters for beta diffusion as follows: $c_0 = 10$, $c_1 = -13$, $S_{hift} = 0.6$, $S_{cale} = 0.39$, $\eta = 10000$, $\omega = 0.99$, and $\pi = 0.95$. We utilize the Adam optimizer with a learning rate of 2e-4. We use EDM's data augmentation approach, but restrict augmented images to a 0-1 range before scaling and shifting. We use the beta diffusion model trained on 200M images to calculate the FID [21]. Explanations regarding the intuition behind these parameter selections can be located in Appendix G.

Below we provide numerical comparison of beta diffusion with not only Gaussian diffusion models but also alternative non-Gaussian or Gaussian-like ones. We also conduct an ablation study to compare KLUB and negative ELBO across different NFE, $\eta$, and $B$. A broad spectrum of diffusion models is encompassed in Table 1, which shows that beta diffusion outperforms all non-Gaussian diffusion models on CIFAR10, including Cold Diffusion [5] and Inverse Heat Dispersion [52], as well as categorical and count-based diffusion models [2, 8, 10]. In comparison to Gaussian diffusion and Gaussian+blurring diffusion, beta diffusion surpasses VDM [35], Soft Diffusion [12], and Blurring Diffusion [25]. While it may fall slightly short of DDPM [23], improved DDPM [46], TPDM+ [78], VP-EDM [34], it remains a competitive alternative that uses non-Gaussian based diffusion.

Table 1: Comparison of the FID scores of various diffusion models trained on CIFAR-10.

| Diffusion Space | Model | FID ($\downarrow$) |
|---|---|---|
| Gaussian | DDPM [23] | 3.17 |
| | VDM [35] | 4.00 |
| | Improved DDPM [46] | 2.90 |
| | TDPM+ [78] | 2.83 |
| | VP-EDM [34] | **1.97** |
| Gaussian+Blurring | Soft Diffusion [12] | 3.86 |
| | Blurring Diffusion [25] | 3.17 |
| Deterministic | Cold Diffusion (image reconstruction) [5] | 80.08 (deblurring) 8.92 (inpainting) |
| | Inverse Heat Dispersion [52] | 18.96 |
| Categorical | D3PM Gauss+Logistic [2] | 7.34 |
| | $\tau$LDR-10 [8] | 3.74 |
| Count | JUMP (Poisson Diffusion) [10] | 4.80 |
| Range-bounded | Beta Diffusion | 3.06 |

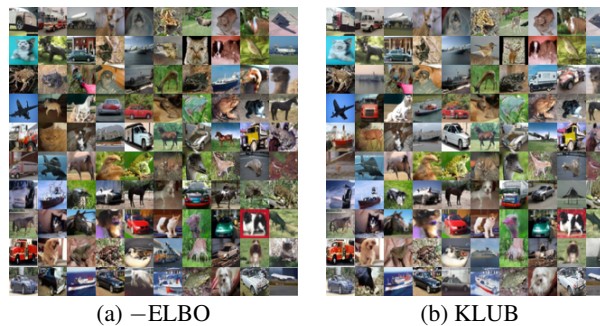

Table 2: Comparing FID scores for KLUB and negative ELBO-optimized Beta Diffusion on CIFAR-10 with varying NFE under $\eta = 10000$ and two different mini-batch sizes $B$.

| Loss $B$ | $-$ELBO 512 | $-$ELBO 288 | KLUB 512 | KLUB 288 |
|---|---|---|---|---|
| 20 | 16.04 | 16.10 | 17.06 | 16.09 |
| 50 | 6.82 | 6.82 | 6.48 | 5.96 |
| 200 | 4.55 | 4.84 | 3.69 | 3.31 |
| 500 | 4.39 | 4.65 | 3.45 | 3.10 |
| 1000 | 4.41 | 4.61 | 3.38 | 3.08 |
| 2000 | 4.50 | 4.66 | 3.37 | **3.06** |

(a) $-$ELBO  (b) KLUB

Figure 4: Uncurated randomly-generated images by beta diffusion optimized with $-$ELBO or KLUB with $\eta = 10000$ and $B = 288$. The generation with NFE $= 1000$ starts from the same random seed.

Table 2 presents a comparison between KLUB and negative ELBO-optimized beta diffusion across different NFE under two different mini-batch sizes $B$. Table 3 in the Appendix includes the results under several different combinations of $\eta$ and $B$. We also include Figure 4 to visually compare generated images under KLUB and negative ELBO. The findings presented in Tables 2 and 3 and Figure 4 provide further validation of KLUB's efficacy in optimizing beta diffusion.

As each training run takes a long time and FID evaluation is also time-consuming, we have not yet optimized the combination of these hyperparameters given the limit of our current computation resources. Thus the results reported in this paper, while demonstrating that beta diffusion can provide competitive image generation performance, do not yet reflect the full potential of beta diffusion. These results are likely to be further improved given an optimized hyperparameter setting or a network architecture that is tailored to beta diffusion. We leave these further investigations to our future work.

## 5 Conclusion

We introduce beta diffusion characterized by the following properties: 1) **Analytic Marginal:** Given a probability-valued data observation $x_0 \in (0, 1)$, the distribution at any time point $t \in [0, 1]$ of the forward beta diffusion chain, expressed as $q(z_t \,|\, x_0)$, is a beta distribution. 2) **Analytical Conditional:** Conditioning on a data $x_0$ and a forward-sampled latent variable $z_t \sim q(z_t \,|\, x_0)$, the forward beta diffusion chain can be reversed from time $t$ to the latent variable at any previous time $s \in [0, t)$ by sampling from an analytic conditional posterior $z_s \sim q(z_s \,|\, z_t, x_0)$ that follows a scaled and shifted beta distribution. 3) **KLUBs:** We introduce the combination of two different Kullback–Leibler Upper Bounds (KLUBs) for optimization and represent them under the log-beta Bregman divergence, showing that their optimal solutions of the generator are both achieved at $f_{\theta^*}(z_t, t) = \mathbb{E}[x_0 \,|\, z_t]$. We also establish the connection between augment-swapped KLUBs and (weighted) negative ELBOs for diffusion models. Our experimental results confirm the distinctive qualities of beta diffusion when applied to generative modeling of range-bounded data spanning disjoint regions or residing in high-dimensional spaces, as well as the effectiveness of KLUBs for optimizing beta diffusion.

## Acknowledgements

The authors acknowledge the support of NSF-IIS 2212418, NIH-R37 CA271186, the Fall 2022 McCombs REG award, the Texas Advanced Computing Center (TACC), and the NSF AI Institute for Foundations of Machine Learning (IFML).

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

# Beta Diffusion: Appendix

---

**Algorithm 1** Training of Beta Diffusion

---

**Require:** Dataset $\mathcal{D}$ whose values are bounded from 0 to 1, Mini-batch size $B$, concentration
parameter $\eta = 10000$, data shifting parameter $S_{hift} = 0.6$, data scaling parameter $S_{cale} = 0.39$,
generator $f_\theta$, loss balance coefficient $\omega = 0.99$, time reversal coefficient $\pi = 0.95$, and a
decreasing function that returns scheduling parameter $\alpha_t \in (0, 1)$ given $t \in [0, 1]$, such as a beta
linear schedule defined by $\alpha_t = e^{-0.5(\beta_{\max} - \beta_{\min})t^2 - \beta_{\min}t}$, where $\beta_{\max} = 20$ and $\beta_{\min} = 0.1$,
and a sigmoid schedule defined by $\alpha_t = 1/(1 + e^{-c_0 - (c_1 - c_0)t})$, where $c_0 = 10$ and $c_1 = -13$.

1: **repeat**
2:     Draw a mini-batch $X_0 = \{x_0^{(i)}\}_{i=1}^B$ from $\mathcal{D}$
3:     **for** $i = 1$ to $B$ **do**                                        ▷ can be run in parallel
4:         $t_i \sim \text{Unif}(1\text{e-}5, 1)$
5:         $s_i = \pi t_i$
6:         Compute $\alpha_{s_i}$ and $\alpha_{t_i}$
7:         $x_0^{(i)} = x_0^{(i)} * S_{cale} + S_{hift}$
8:         $z_{t_i} \sim \text{Beta}(\eta \alpha_{t_i} x_0^{(i)}, \eta(1 - \alpha_{t_i} x_0^{(i)}))$
9:         $\hat{x}_0^{(i)} = f_\theta(z_{t_i}, t_i) * S_{cale} + S_{hift}$
10:        Using (18) to compute the loss as

$$
\begin{aligned}
\mathcal{L}_i &= \omega \text{KLUB}(s_i, z_{t_i}, x_0^{(i)}) + (1 - \omega)\text{KLUB}(z_{t_i}, x_0^{(i)}) \\
&= \omega D_{\ln B(a,b)} \Big\{ \Big[\eta(\alpha_{s_i} - \alpha_{t_i})x_0^{(i)}, \eta(1 - \alpha_{s_i}x_0^{(i)})\Big], \ \Big[\eta(\alpha_{s_i} - \alpha_{t_i})\hat{x}_0^{(i)}, \eta(1 - \alpha_{s_i}\hat{x}_0^{(i)})\Big] \Big\} \\
&\quad + (1 - \omega)D_{\ln B(a,b)} \Big\{ \Big[\eta\alpha_{t_i}x_0^{(i)}, \eta(1 - \alpha_{t_i}x_0^{(i)})\Big], \ \Big[\eta\alpha_{t_i}\hat{x}_0^{(i)}, \eta(1 - \alpha_{t_i}\hat{x}_0^{(i)})\Big] \Big\} \quad (24)
\end{aligned}
$$

       or swap the two arguments of both log-beta Bregman divergences in (24) if the loss is (weighted)
       $-$ELBO.
11:     **end for**
12:     Perform SGD with $\frac{1}{B}\nabla_\theta \sum_{i=1}^B \mathcal{L}_i$
13: **until** converge

---

Table 3: Comparing FID scores for KLUB and negative ELBO-optimized Beta Diffusion on the
CIFAR-10 image dataset with varying NFE under several different combinations of concentration
parameter $\eta$ and mini-batch size $B$. We train the model with 200M images and use it to calculate FID.
We compute FID one time in each experiment. The other model parameters are set as $S_{cale} = 0.39$,
$S_{hift} = 0.60$, $\pi = 0.95$, $\omega = 0.99$, $lr = 2\text{e-}4$, $c_0 = 10$, and $c_1 = -13$.

| Loss | $\eta \times 10^{-4}$ | $B$ | NFE = 10 | 20 | 50 | 200 | 500 | 1000 | 2000 |
|------|------|-----|------|------|------|------|------|------|------|
| $-$ELBO | 1 | 512 | 37.64 | 16.04 | 6.82 | 4.55 | 4.39 | 4.41 | 4.50 |
| $-$ELBO | 1 | 288 | 37.54 | 16.10 | 6.82 | 4.84 | 4.65 | 4.61 | 4.66 |
| KLUB | 1 | 512 | 39.05 | 17.06 | 6.48 | 3.69 | 3.45 | 3.38 | 3.37 |
| KLUB | 0.1 | 512 | 41.28 | 20.38 | 9.72 | 5.85 | 4.98 | 4.90 | 4.88 |
| KLUB | 0.3 | 512 | 36.70 | 16.47 | 7.03 | 4.10 | 3.65 | 3.67 | 3.66 |
| KLUB | 1 | 288 | 37.67 | 16.09 | 5.96 | 3.31 | 3.10 | 3.08 | **3.06** |
| KLUB | 0.8 | 288 | 36.46 | 15.58 | 5.98 | 3.49 | 3.22 | 3.21 | 3.23 |
| KLUB | 1.2 | 288 | 38.49 | 16.59 | 6.31 | 3.68 | 3.36 | 3.37 | 3.24 |
| KLUB | 1 | 128 | 38.16 | 16.47 | 6.29 | 3.74 | 3.44 | 3.40 | 3.47 |
| KLUB | 2 | 128 | 39.80 | 17.09 | 6.58 | 3.93 | 3.76 | 3.75 | 3.65 |

**Algorithm 2** Sampling of Beta Diffusion

---

**Require:** Number of function evaluations (NFE) $J = 200$, generator $f_\theta$, and timesteps $\{t_j\}_{j=0}^J$:

$t_j = 1 - (1 - 1\text{e-}5) * (J - j)/(J - 1)$ for $j = 1, \ldots, J$ and $t_0 = 0$

1: **if** NFE $> 350$ **then**
2: $\quad \alpha_{t_j} = 1/(1 + e^{-c_0 - (c_1 - c_0)t_j})$
3: **else**
4: $\quad \alpha_{t_j} = (1/(1 + e^{-c_1}))^{t_j}$
5: **end if**
6: Initialize $\hat{x}_0 = \mathbb{E}[x_0] * S_{cale} + S_{hift}$
7: $z_{t_J} \sim \text{Beta}(\eta \alpha_{t_J} \hat{x}_0, \eta(1 - \alpha_{t_J} \hat{x}_0))$
8: **for** $j = J$ to 1 **do**
9: $\quad \hat{x}_0 = f_\theta(z_{t_j}, \alpha_{t_j}) * S_{cale} + S_{hift}$
10: $\quad p_{(t_{j-1} \leftarrow t_j)} \sim \text{Beta}\left(\eta(\alpha_{t_{j-1}} - \alpha_{t_j})\hat{x}_0, \eta(1 - \alpha_{t_{j-1}}\hat{x}_0)\right),$

$\qquad \triangleright$ which is implemented in the logit space as

$$\text{logit}(p_{(t_{j-1} \leftarrow t_j)}) = \ln u - \ln v,$$
$$u \sim \text{Gamma}(\eta(\alpha_{t_{j-1}} - \alpha_{t_j})\hat{x}_0, 1),$$
$$v \sim \text{Gamma}(\eta(1 - \alpha_{t_{j-1}}\hat{x}_0), 1)$$

11: $\quad z_{t_{j-1}} = z_{t_j} + (1 - z_{t_j})p_{(t_{j-1} \leftarrow t_j)},$

$\qquad \triangleright$ which is implemented in the logit space as

$$\text{logit}(z_{t_{j-1}}) = \ln\left(e^{\text{logit}(z_{t_j})} + e^{\text{logit}(p_{(t_{j-1} \leftarrow t_j)})} + e^{\text{logit}(z_{t_j}) + \text{logit}(p_{(t_{j-1} \leftarrow t_j)})}\right)$$

12: **end for**
13: **return** $(\hat{x}_0 - S_{hift})/S_{cale}$ or $(z_{t_0}/\alpha_{t_0} - S_{hift})/S_{cale}$

---

## A   Log-beta Divergence and KL Divergence between Beta Distributions

By the definition of Bregman divergence, the log-beta divergence corresponding to the log-beta function $\ln B(a, b) = \ln \Gamma(a) + \ln \Gamma(b) - \ln \Gamma(a + b)$, which is differentiable, and strictly convex on $(0, \infty)^2$ as a function of $a$ and $b$, can be expressed as

$$
\begin{aligned}
&D_{\ln B(a,b)}((\alpha_q, \beta_q), (\alpha_p, \beta_p)) \\
&= \ln \frac{B(\alpha_q, \beta_q)}{B(\alpha_p, \beta_p)} - (\alpha_q - \alpha_p, \beta_q - \beta_p) \begin{pmatrix} \nabla_{\alpha_p} \ln B(\alpha_p, \beta_p) \\ \nabla_{\beta_p} \ln B(\alpha_p, \beta_p) \end{pmatrix} \\
&= \ln \frac{B(\alpha_q, \beta_q)}{B(\alpha_p, \beta_p)} - (\alpha_q - \alpha_p, \beta_q - \beta_p) \begin{pmatrix} \psi(\alpha_p) - \psi(\alpha_p + \beta_p) \\ \psi(\beta_p) - \psi(\alpha_p + \beta_p) \end{pmatrix} \\
&= \ln \frac{B(\alpha_q, \beta_q)}{B(\alpha_p, \beta_p)} - (\alpha_q - \alpha_p)\psi(\alpha_p) - (\beta_q - \beta_p)\psi(\beta_p) + (\alpha_q - \alpha_p + \beta_q - \beta_p)\psi(\alpha_p + \beta_p),
\end{aligned}
$$

which is equivalent to the analytic expression of

$$\text{KL}(\text{Beta}(\alpha_p, \beta_p) || \text{Beta}(\alpha_q, \beta_q)).$$

## B Proof

*Proof of Lemma 1.* The joint distribution of $z_t$ and $z_s$ in (5) can be expressed as

$$q(z_t, z_s \,|\, x_0) = q(z_t \,|\, z_s)q(z_s \,|\, x_0)$$
$$= \frac{1}{z_s}\text{Beta}\left(\frac{z_t}{z_s}; \eta\alpha_t x_0, \eta(\alpha_s - \alpha_t)x_0\right)\text{Beta}(z_s; \eta\alpha_s x_0, \eta(1 - \alpha_s x_0))$$
$$= \frac{1}{z_s}\frac{\Gamma(\eta\alpha_s x_0)}{\Gamma(\eta\alpha_t x_0)\Gamma(\eta(\alpha_s - \alpha_t)x_0)}\left(\frac{z_t}{z_s}\right)^{\eta\alpha_t x_0 - 1}\left(1 - \frac{z_t}{z_s}\right)^{\eta(\alpha_s - \alpha_t)x_0 - 1}$$
$$\times \frac{\Gamma(\eta)}{\Gamma(\eta\alpha_s x_0)\Gamma(\eta(1 - \alpha_s x_0))}z_s^{\eta\alpha_s x_0 - 1}(1 - z_s)^{\eta(1 - \alpha_s x_0) - 1}$$
$$= \frac{\Gamma(\eta)}{\Gamma(\eta\alpha_t x_0)\Gamma(\eta(\alpha_s - \alpha_t)x_0)\Gamma(\eta(1 - \alpha_s x_0))}$$
$$\times z_t^{\eta\alpha_t x_0 - 1}(z_s - z_t)^{\eta(\alpha_s - \alpha_t)x_0 - 1}(1 - z_s)^{\eta(1 - \alpha_s x_0) - 1}. \tag{25}$$

The joint distribution of $z_t$ and $z_s$ in (6) can be expressed as

$$q(z_t, z_s \,|\, x_0) = q(z_s \,|\, z_t, x_0)q(z_t \,|\, x_0)$$
$$= \frac{1}{1 - z_t}\text{Beta}\left(\frac{z_s - z_t}{1 - z_t}; \eta(\alpha_s - \alpha_t)x_0, \eta(1 - \alpha_s x_0)\right)\text{Beta}(z_t; \eta\alpha_t x_0, \eta(1 - \alpha_t x_0))$$
$$= \frac{1}{1 - z_t}\frac{\Gamma(\eta(1 - \alpha_t x_0))}{\Gamma(\eta(1 - \alpha_s x_0))\Gamma(\eta(\alpha_s - \alpha_t)x_0)}$$
$$\times \left(\frac{z_s - z_t}{1 - z_t}\right)^{\eta(\alpha_s - \alpha_t)x_0 - 1}\left(1 - \frac{z_s - z_t}{1 - z_t}\right)^{\eta(1 - \alpha_s x_0) - 1}$$
$$\times \frac{\Gamma(\eta)}{\Gamma(\eta\alpha_t x_0)\Gamma(\eta(1 - \alpha_t x_0))}z_t^{\eta\alpha_t x_0 - 1}(1 - z_t)^{\eta(1 - \alpha_t x_0) - 1}$$
$$= \frac{\Gamma(\eta)}{\Gamma(\eta\alpha_t x_0)\Gamma(\eta(\alpha_s - \alpha_t)x_0)\Gamma(\eta(1 - \alpha_s x_0))}$$
$$\times (z_s - z_t)^{\eta(\alpha_s - \alpha_t)x_0 - 1}(1 - z_s)^{\eta(1 - \alpha_s x_0) - 1}z_t^{\eta\alpha_t x_0 - 1}. \tag{26}$$

The joint distribution shown in (25) is the same as that in (26). $\square$

*Proof of Lemma 2.* Since we can re-express $q(z_t)$ and $q(z_s, z_t)$ as

$$q(z_t) = \mathbb{E}_{x_0 \sim p_{data}(x_0)}[q(z_t \,|\, x_0)],$$

$$q(z_s, z_t) = \mathbb{E}_{x_0 \sim p_{data}(x_0)}[q(z_s, z_t \,|\, x_0)],$$

and $q(z_s, z_t \,|\, x_0) = q(z_s \,|\, z_t, x_0)q(z_t \,|\, x_0)$, using the convexity of the KL divergence, we have

$$\text{KL}(p_\theta(z_s \,|\, z_t)q(z_t)||q(z_s, z_t))$$
$$= \text{KL}(\mathbb{E}_{x_0 \sim p_{data}(x_0)}[p_\theta(z_s \,|\, z_t)q(z_t \,|\, x_0)]||\mathbb{E}_{x_0 \sim p_{data}(x_0)}[q(z_s, z_t \,|\, x_0)])$$
$$\leq \mathbb{E}_{x_0 \sim p_{data}(x_0)}[\text{KL}(p_\theta(z_s \,|\, z_t)q(z_t \,|\, x_0)||q(z_s, z_t \,|\, x_0))]$$
$$= \mathbb{E}_{x_0 \sim p_{data}(x_0)}\mathbb{E}_{(z_s, z_t) \sim p_\theta(z_s \,|\, z_t)q(z_t \,|\, x_0)}\ln\frac{p_\theta(z_s \,|\, z_t)q(z_t \,|\, x_0)}{q(z_s \,|\, z_t, x_0)q(z_t \,|\, x_0)}$$
$$= \mathbb{E}_{x_0 \sim p_{data}(x_0)}\mathbb{E}_{z_t \sim q(z_t \,|\, x_0)}\mathbb{E}_{z_s \sim p_\theta(z_s \,|\, z_t)}\ln\frac{p_\theta(z_s \,|\, z_t)}{q(z_s \,|\, z_t, x_0)}$$
$$= \mathbb{E}_{(z_t, x_0) \sim q(z_t \,|\, x_0)p_{data}(x_0)}[\text{KL}(p_\theta(z_s \,|\, z_t)||q(z_s \,|\, z_t, x_0))]$$
$$= \mathbb{E}_{(z_t, x_0) \sim q(z_t \,|\, x_0)p_{data}(x_0)}[\text{KL}(q(z_s \,|\, z_t, \hat{x}_0 = f_\theta(z_t, t))||q(z_s \,|\, z_t, x_0))].$$

$\square$

*Proof of Lemma 4.* Since we can re-express $p_\theta(z_t')$ and $q(z_t')$ as

$$p_\theta(z_t') := \mathbb{E}_{z_t \sim q(z_t)}[q(z_t' \mid \hat{x}_0 = f_\theta(z_t, t))] = \mathbb{E}_{(z_t, x_0) \sim q(z_t \mid x_0) p_{data}(x_0)}[q(z_t' \mid \hat{x}_0 = f_\theta(z_t, t))],$$

$$q(z_t') = \mathbb{E}_{(z_t, x_0) \sim q(z_t \mid x_0) p_{data}(x_0)}[q(z_t' \mid x_0)],$$

using the convexity of the KL divergence, we have

$$\mathrm{KL}(p_\theta(z_t') \| q(z_t'))$$
$$= \mathrm{KL}(\mathbb{E}_{(z_t, x_0) \sim q(z_t \mid x_0) p_{data}(x_0)}[q(z_t' \mid \hat{x}_0 = f_\theta(z_t, t))] \| \mathbb{E}_{(z_t, x_0) \sim q(z_t \mid x_0) p_{data}(x_0)}[q(z_t' \mid x_0)])$$
$$\leq \mathbb{E}_{(z_t, x_0) \sim q(z_t \mid x_0) p_{data}(x_0)}[\mathrm{KL}(q(z_t' \mid f_\theta(z_t, t)) \| q(z_t' \mid x_0))].$$

$\square$

*Proof of Lemma 6.* The proof utilizes the convexity of the KL divergence to show that

$$\mathrm{KL}(p_\theta(z_{t_{1:T}}) \| q(z_{t_{1:T}})) = \mathrm{KL}(\mathbb{E}_{x_0 \sim p_{data}(x_0)}[p_\theta(z_{1:T} \mid x_0)] \| \mathbb{E}_{x_0 \sim p_{data}(x_0)}[q(z_{1:T} \mid z_0)])$$
$$\leq \mathbb{E}_{x_0 \sim p_{data}(x_0)} \mathrm{KL}(p_\theta(z_{t_{1:T}} \mid x_0) \| q(z_{t_{1:T}} \mid x_0))$$
$$= \mathbb{E}_{x_0 \sim p_{data}(x_0)} \mathbb{E}_{z_{t_{1:T}} \sim p_\theta(z_{t_{1:T}} \mid x_0)} \ln \frac{p_\theta(z_{t_{1:T}} \mid x_0)}{q(z_{t_{1:T}} \mid x_0)}$$
$$= \mathbb{E}_{x_0 \sim p_{data}(x_0)} \Bigg[ \mathrm{KL}(p_{prior}(z_{t_T} \mid x_0) \| q(z_{t_T} \mid x_0))$$
$$+ \sum_{j=2}^{T} \mathbb{E}_{z_{t_j} \sim p_\theta(z_{t_j} \mid x_0)} \mathbb{E}_{z_{t_{j-1}} \sim p_\theta(z_{t_{j-1}} \mid z_{t_j})} \ln \frac{p_\theta(z_{t_{j-1}} \mid z_{t_j})}{q(z_{t_{j-1}} \mid z_{t_j}, x_0)} \Bigg].$$

We note $p_\theta(z_{t_j} \mid x_0) \propto p(x_0 \mid z_{t_j}) p_\theta(z_{t_j})$, which in general is intractable to sample from, motivating us to replace $p_\theta(z_{t_j} \mid x_0)$ with $q(z_{t_j} \mid x_0)$ for tractable computation. $\square$

*Proof of Lemma 7.* Utilizing the convexity of the KL divergence, we present an upper-bound of the augmented-swapped KL divergence $\mathrm{KL}(q(z_{t_{1:T}}) \| p_\theta(z_{t_{1:T}}))$, referred to as AS-KLUB, as

$$\mathrm{KL}(q(z_{t_{1:T}}) \| p_\theta(z_{t_{1:T}})) = \mathrm{KL}(\mathbb{E}_{x_0 \sim p_{data}(x_0)}[q(z_{t_{1:T}} \mid x_0)] \| \mathbb{E}_{x_0 \sim p_{data}(x_0)}[p_\theta(z_{t_{1:T}} \mid x_0)])$$
$$\leq \text{AS-KLUB} = \mathbb{E}_{x_0 \sim p_{data}(x_0)} \mathrm{KL}(q(z_{t_{1:T}} \mid x_0) \| p_\theta(z_{t_{1:T}} \mid x_0))$$
$$= \mathbb{E}_{x_0 \sim p_{data}(x_0)} \mathbb{E}_{z_{t_{1:T}} \sim q(z_{t_{1:T}} \mid x_0)} \ln \frac{q(z_{t_{1:T}} \mid x_0)}{p_\theta(z_{t_{1:T}} \mid x_0)}$$
$$= \mathbb{E}_{x_0 \sim p_{data}(x_0)} \Bigg[ \mathrm{KL}(q(z_{t_T} \mid x_0) \| p_{prior}(z_{t_T} \mid x_0))$$
$$+ \sum_{j=2}^{T} \mathbb{E}_{z_{t_j} \sim q(z_{t_j} \mid x_0)} \mathbb{E}_{z_{t_{j-1}} \sim q(z_{t_{j-1}} \mid z_{t_j}, x_0)} \ln \frac{q(z_{t_{j-1}} \mid z_{t_j}, x_0)}{p_\theta(z_{t_{j-1}} \mid z_{t_j})} \Bigg].$$

Utilizing the convexity of the negative logarithmic function, we have

$$- \mathbb{E}_{x_0 \sim p_{data}(x_0)} \ln p_\theta(x_0) = -\mathbb{E}_{x_0 \sim p_{data}(x_0)} \ln \mathbb{E}_{p_\theta(z_{t_{1:T}})}[p(x_0 \mid z_{t_{1:T}})]$$
$$= -\mathbb{E}_{x_0 \sim p_{data}(x_0)} \ln \mathbb{E}_{q(z_{t_{1:T}} \mid x_0)} \left[ \frac{p(x_0 \mid z_{t_{1:T}}) p_\theta(z_{t_{1:T}})}{q(z_{t_{1:T}} \mid x_0)} \right]$$
$$\leq -\text{ELBO} = -\mathbb{E}_{x_0 \sim p_{data}(x_0)} \mathbb{E}_{q(z_{t_{1:T}} \mid x_0)} \left[ \ln \frac{p(x_0 \mid z_{t_{1:T}}) p_\theta(z_{t_{1:T}})}{q(z_{t_{1:T}} \mid x_0)} \right]$$
$$= \mathbb{E}_{x_0 \sim p_{data}(x_0)} \Bigg[ - \mathbb{E}_{q(z_{t_1} \mid x_0)} \ln p(x_0 \mid z_{t_1}) + \mathrm{KL}[q(z_{t_T} \mid x_0) \| p_{prior}(z_{t_T})]$$
$$+ \sum_{j=2}^{T} \mathbb{E}_{z_{t_j} \sim q(z_{t_j} \mid x_0)} \mathbb{E}_{z_{t_{j-1}} \sim q(z_{t_{j-1}} \mid z_{t_j}, x_0)} \ln \frac{q(z_{t_{j-1}} \mid z_{t_j}, x_0)}{p_\theta(z_{t_{j-1}} \mid z_{t_j})} \Bigg].$$

Thus, when we disregard its first term, AS-KLUB is equivalent to $-$ELBO without its first two terms. Since these three terms typically do not affect optimization, optimizing the generator $f_\theta$ with AS-KLUB is equivalent to using $-$ELBO. $\square$

## C Derivation of (Weighted) ELBOs of Gaussian Diffusion from KLUB

Let us denote $\alpha_0 = 1$ and $z_0 = x_0$. Following the definition in Ho et al. [23] and Song et al. [58], we define a Gaussian diffusion-based generative model as

$$z_1 \sim \mathcal{N}(\sqrt{\alpha_1}x_0, 1 - \alpha_1),$$

$$z_2 \sim \mathcal{N}\left(\sqrt{\frac{\alpha_2}{\alpha_1}}z_1, 1 - \frac{\alpha_2}{\alpha_1}\right),$$

$$\ldots$$

$$z_t \sim \mathcal{N}\left(\sqrt{\frac{\alpha_t}{\alpha_{t-1}}}z_{t-1}, 1 - \frac{\alpha_t}{\alpha_{t-1}}\right),$$

$$\ldots$$

$$z_T \sim \mathcal{N}\left(\sqrt{\frac{\alpha_T}{\alpha_{T-1}}}z_{T-1}, 1 - \frac{\alpha_T}{\alpha_{T-1}}\right),$$

where the diffusion scheduling parameters $\alpha_t$ in this paper is related to $\beta_t$ in Ho et al. [23] as

$$\beta_t := 1 - \frac{\alpha_t}{\alpha_{t-1}}.$$

The same as the derivations in Ho et al. [23], we can express the forward marginal distribution at time $t$ as

$$z_t \sim \mathcal{N}(\sqrt{\alpha_t}x_0, 1 - \alpha_t).$$

Assuming $\mathbb{E}[x_0] = 0$ and $\mathrm{var}[x_0] = 1$, the signal-to-noise ratio at time $t$ is defined as

$$\mathrm{SNR}_t = \mathbb{E}_{x_0}\left[\left(\frac{\mathbb{E}[z_t \mid x_0]}{\mathrm{std}[z_t \mid x_0]}\right)^2\right] = \frac{\alpha_t}{1 - \alpha_t}\mathbb{E}[x_0^2] = \frac{\alpha_t}{1 - \alpha_t}.$$

Since $\sqrt{\frac{\alpha_{t-1}}{\alpha_t}}x_t \sim \mathcal{N}(z_{t-1}, \frac{\alpha_{t-1}}{\alpha_t} - 1)$ and $z_{t-1} \sim \mathcal{N}(\sqrt{\alpha_{t-1}}z_0, 1 - \alpha_{t-1})$, using the conjugacy of the Gaussian distributions with respect to their mean, we can express the conditional posterior as $q(z_{t-1} \mid x_0, z_t)$

$$= \mathcal{N}\left(\left(\frac{1}{1 - \alpha_{t-1}} + \frac{\alpha_t}{\alpha_{t-1} - \alpha_t}\right)^{-1}\left(\frac{\sqrt{\alpha_{t-1}}x_0}{1 - \alpha_{t-1}} + \frac{\sqrt{\alpha_{t-1}\alpha_t}x_t}{\alpha_{t-1} - \alpha_t}\right), \left(\frac{1}{1 - \alpha_{t-1}} + \frac{\alpha_t}{\alpha_{t-1} - \alpha_t}\right)^{-1}\right)$$

$$= \mathcal{N}\left(\frac{\sqrt{\alpha_{t-1}}}{1 - \alpha_t}(1 - \frac{\alpha_t}{\alpha_{t-1}})x_0 + \frac{1 - \alpha_{t-1}}{1 - \alpha_t}\sqrt{\frac{\alpha_t}{\alpha_{t-1}}}z_t, \ \frac{1 - \alpha_{t-1}}{1 - \alpha_t}(1 - \frac{\alpha_t}{\alpha_{t-1}})\right).$$

The forward Gaussian diffusion chain

$$q(z_{1:T} \mid x_0) = \prod_{t=1}^{T} p(z_t \mid z_{t-1})$$

can be equivalently sampled in reverse order as

$$q(z_{1:T} \mid x_0) = q(z_T \mid x_0)\prod_{t=2}^{T} q(z_{t-1} \mid z_t, x_0).$$

As this reverse chain is non-causal and non-Markovian, we need to approximate it with a Markov chain during inference, which is expressed as

$$p(z_{1:T}) = p_{prior}(z_T)\prod_{t=2}^{T} p_\theta(z_{t-1} \mid z_t).$$

The usual strategy [58, 35, 72] is to utilize the conditional posterior to define

$$p_\theta(z_{t-1} \mid z_t) = q(z_{t-1} \mid z_t, \hat{x}_0 = f_\theta(z_t, t)).$$

In what follows, we redefine the time $t$ as a continuous variable between 0 and 1. We let $t_{1:T}$ be a set of $T$ discrete time points within that interval, with $0 \le t_1 < t_2 < \ldots < t_T \le 1$. The corrupted data observations over these $T$ time points are defined as $z_{t_{1:T}}$.

## C.1 Optimization of Gaussian Diffusion via Negative ELBO

Viewing $p_\theta(x_0 \mid z_{1:T})$ as the decoder, $p_\theta(z_{1:T})$ as the prior, and $q(z_{1:T} \mid x_0)$ as the inference network, we can optimize the generator parameter via the negative ELBO as

$$- \mathbb{E}_{x_0} \ln p_\theta(x_0) \leq -\text{ELBO} = \mathbb{E}_{x_0} \mathbb{E}_{q(z_{t_{1:T}} \mid x_0)} \left[ -\ln \frac{p(x_0 \mid z_{t_{1:T}}) p(z_{t_{1:T}})}{q(z_{t_{1:T}} \mid x_0)} \right]$$

$$= \mathbb{E}_{x_0} \mathbb{E}_{q(z_{t_1} \mid x_0)} \ln p(x_0 \mid z_{t_1}) + \mathbb{E}_{x_0} \text{KL}[q(z_{t_T} \mid x_0) || p_{prior}(z_{t_T})] + \sum_{j=2}^{T} \mathbb{E}_{x_0} \mathbb{E}_{q(z_{t_j} \mid x_0)} [L(t_{j-1}, z_{t_j}, x_0)],$$

where the first two terms are often ignored and the focus is placed on the remaining $T - 2$ terms, defined as

$$L(s, z_t, x_0) = \text{KL}(q(z_s \mid z_t, x_0) || q(z_s \mid z_t, \hat{x}_0 = f_\theta(z_t, t)))$$

$$= \frac{1}{2 \frac{1-\alpha_s}{1-\alpha_t}(1 - \frac{\alpha_t}{\alpha_s})} \left( \frac{\sqrt{\alpha_s}}{1 - \alpha_t} \left( 1 - \frac{\alpha_t}{\alpha_s} \right) \right)^2 \|x_0 - f_\theta(z_t, t)\|_2^2$$

$$= \frac{1}{2} \left( \frac{\alpha_s}{1 - \alpha_s} - \frac{\alpha_t}{1 - \alpha_t} \right) \|x_0 - f_\theta(z_t, t)\|_2^2, \tag{27}$$

where $0 < s < t < 1$. We choose $s = \max(t - 1/T, 0)$ during training.

Since $z_t = \sqrt{\alpha_t} x_0 + \sqrt{1 - \alpha_t} \epsilon_t$, $\epsilon_t \sim \mathcal{N}(0, 1)$, it is true that

$$x_0 = \frac{z_t - \sqrt{1 - \alpha_t} \epsilon_t}{\sqrt{\alpha_t}}.$$

Instead of directly predicting $x_0$ given $x_t$, we can equivalently predict $\hat{\epsilon}_t = \epsilon_\theta(z_t, t)$ given $z_t$ and let

$$\hat{x}_0 = f_\theta(z_t, t) = \frac{z_t - \sqrt{1 - \alpha_t} \epsilon_\theta(z_t, t)}{\sqrt{\alpha_t}}.$$

Thus (27) can also be written as

$$L(s, z_t, x_0) = \frac{1}{2} \left( \frac{\alpha_s}{1 - \alpha_s} - \frac{\alpha_t}{1 - \alpha_t} \right) \|x_0 - f_\theta(z_t, t)\|_2^2$$

$$= \frac{1}{2} (\text{SNR}_s - \text{SNR}_t) \|x_0 - f_\theta(z_t, t)\|_2^2$$

$$= \frac{1}{2} \left( \frac{\alpha_s}{1 - \alpha_s} - \frac{\alpha_t}{1 - \alpha_t} \right) \frac{1 - \alpha_t}{\alpha_t} \|\epsilon_t - \epsilon_\theta(z_t, t)\|_2^2$$

$$= \frac{1}{2} \left( \frac{\text{SNR}_s}{\text{SNR}_t} - 1 \right) \|\epsilon_t - \epsilon_\theta(z_t, t)\|_2^2, \tag{28}$$

which agrees with Equations 13 and 14 in Kingma et al. [35].

## C.2 Optimization of Gaussian Diffusion via KLUB Conditional

Following the definition of KLUB (Conditional), for Gaussian diffusion we have

$$\text{KLUB}(s, z_t, x_0) = \text{KL}(q(z_s \mid \hat{x}_0 = f_\theta(z_t, t), z_t) || q(z_s \mid x_0, z_t))$$

For two Gaussian distributions $q_1$ and $q_2$ that have the same variance, we have $\text{KL}(q_1 || q_2) = \text{KL}(q_2 || q_1)$ and hence

$$\text{KLUB}(s, z_t, x_0) = \text{KL}(q(z_s \mid \hat{x}_0 = f_\theta(z_t, t), z_t) || q(z_s \mid x_0, z_t))$$

$$= \text{KL}(q(z_s \mid x_0, z_t) || q(z_s \mid \hat{x}_0 = f_\theta(z_t, t), z_t))$$

$$= L(s, z_t, x_0).$$

Therefore, over the same set of time points $t_{1:T}$, optimizing via KLUB conditional is identical to optimizing via $-$ELBO. As analyzed before, optimizing via KLUB conditional (*i.e.*, $-$ELBO) may not be able to directly counteract the error accumulated over the course of diffusion, and hence could lead to slow convergence.

## C.3 Optimization of Gaussian Diffusion via KLUB Marginal and SNR-weighted Negative ELBO

Following the definition of KLUB (Marginal), for Gaussian diffusion we have

$$\begin{aligned}
\text{KLUB}(z_t, x_0) &= \text{KL}(\mathcal{N}(\sqrt{\alpha_t}x_0, 1-\alpha_t)||\mathcal{N}(\sqrt{\alpha_t}f(z_t,t), 1-\alpha_t) \\
&= \frac{\alpha_t}{1-\alpha_t}\|x_0 - f(z_t,t)\|_2^2 \\
&= \text{SNR}_t\|x_0 - f(z_t,t)\|_2^2 \\
&= \|\epsilon_0 - \epsilon_\theta(z_t,t)\|_2^2.
\end{aligned}$$

It is surprising to find out that the KLUB Marginal is identical to the SNR weighted $-$ELBO first introduced in Ho et al. [23] and further discussed in Hang et al. [20].

## C.4 Optimization of Gaussian Diffusion via KLUB

Following beta diffusion, we can also combine two KLUBs as

$$\begin{aligned}
&\omega\text{KLUB}(s, z_t, x_0) + (1-\omega)\text{KLUB}(z_t, x_0) \\
&= \left[\frac{\omega}{2}\left(\frac{\alpha_s}{1-\alpha_s}\frac{1-\alpha_t}{\alpha_t} - 1\right) + (1-\omega)\right]\|\epsilon_0 - \epsilon_\theta(z_t,t)\|_2^2.
\end{aligned}$$

Since $\frac{\alpha_s}{1-\alpha_s}\frac{1-\alpha_t}{\alpha_t}$ is in general close to 1 when $s$ is not too far from $t$, when $\omega$ is not too close to 1, a combination of these two KLUBs does not result in an algorithm that clearly differs from the use of an SNR-weighted negative ELBO.

# D Illustration of Forward and Reverse Beta Diffusion

**Illustration of Forward Beta Diffusion.** We first visualize the beta forward diffusion process by displaying a true image $x_0$ and its noise-corrupted versions over the course of the forward diffusion process. Specifically, we display the noise corrupted and masked image at time $t = 0, 0.05, 0.1\ldots, 1$ as

$$\tilde{z}_t = \max\left(\min\left(\frac{1}{S_{cale}}\left(\frac{z_t}{\alpha_t} - S_{hift}\right), 1\right), 0\right), \ z_t \sim \text{Beta}(\eta\alpha_t x_0, \eta(1-\alpha_t x_0)).$$

It is clear from Figure 1 that with multiplicative beta-distributed noises, the image becomes both noisier and sparser as time increases, and eventually becomes almost completely dark. Thus the forward process of beta diffusion can be considered as simultaneously noising and masking the pixels. This clearly differs beta diffusion from Gaussian diffusion, whose forward diffusion gradually applies additive random noise and eventually ends at a Gaussian random noise. In addition, the reverse diffusion process of Gaussian diffusion can be considered a denoising process, whereas that of beta diffusion is simultaneously demasking and denoising the data, as illustrated below.

**Illustration of Reverse Beta Diffusion.** We further illustrate the reverse process of beta diffusion by displaying

$$\tilde{z}_{t_{j-1}} = \max\left(\min\left(\frac{1}{S_{cale}}\left(\frac{z_{t_{j-1}}}{\alpha_{t_{j-1}}} - S_{hift}\right), 1\right), 0\right),$$

where $z_{t_{j-1}} = z_{t_j} + (1 - z_{t_j})p_{(t_{j-1}\leftarrow t_j)}$ is iteratively computed as in Algorithm 2. We also display

$$\hat{x}_0 = f_\theta(z_{t_j}, t_j) \approx \mathbb{E}[x_0 \,|\, z_{t_j}],$$

where the approximation would become more and more accurate when $\theta$ approaches its theoretical optimal $\theta^*$, under which we have $f_{\theta^*}(z_{t_j}, t_j) = \mathbb{E}[x_0 \,|\, z_{t_j}]$.

As shown in Figure 2, starting from a random image drawn from

$$z_{t_J} \sim \text{Beta}(\eta\alpha_{t_J}\hat{x}_0, \eta(1-\alpha_{t_J}\hat{x}_0)), \ \hat{x}_0 = \mathbb{E}[x_0],$$

most of whose pixels would be completely dark, beta diffusion gradually demasks and denoises the image towards a clean image through multiplicative transforms, as shown in Algorithm 2.

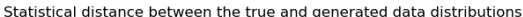

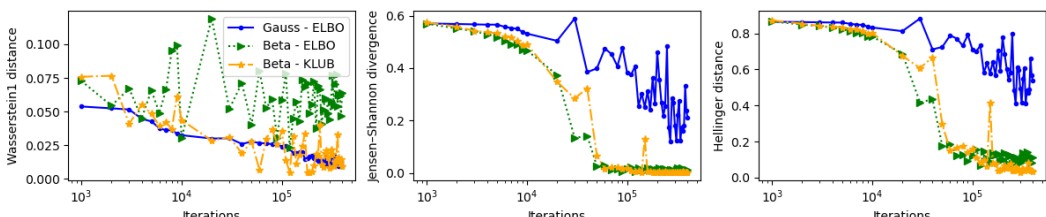

Figure 5: Comparison of the statistical distances between the true and generated data distributions over the course of training. The blue, green, and orange curves are for "Gauss ELBO," "Beta ELBO," and "Beta KLUB," respectively. From the left to right are the plots for Wasserstein-1 distance, Jensen–Shannon divergence, and Hellinger distance, respectively.

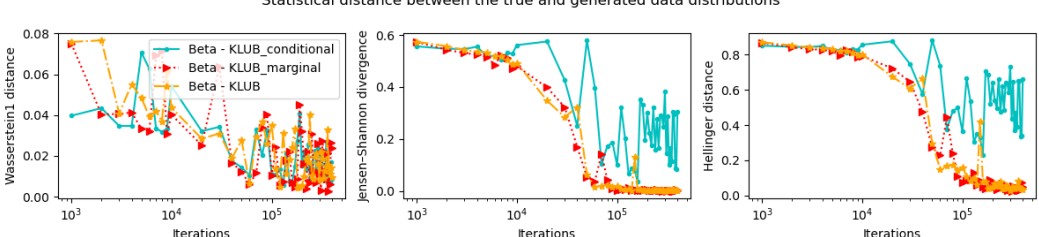

Figure 6: Analogy to Figure 5 for comparing "KLUB Conditional," "KLUB Marginal," and "KLUB."

# E   Additional Results for Synthetic Data

**Quantitative Evaluation Metrics.**  We consider Wasserstein-1, Jensen–Shannon divergence (JSD), and Hellinger distance as three complimentary evaluation metrics. Wasserstein-1 has a high tolerance of misalignment between the supports of the true data density and these of the generated data. In other words, it is not sensitive to both misaligning the modes of the true density and these of the generated density and placing density in the regions where there are no data. By contrast, both JSD and Hellinger can well reflect the misalignments between the supports of high data density regions.

*Wasserstein-1*: We monitor the performance of different algorithms during training by generating 100k data points using the trained model and drawing 100k data points from the data distribution, and computing the *Wasserstein-1* distance between their empirical distributions, which can be done by sorting them and taking the mean of their element-wise absolute differences. If the true distribution is discrete in that it takes values uniformly at random from a discrete set $\mathcal{D}$, then we compute the Wasserstein-1 distance by sorting a set of 10k generated samples from small to large and calculating the mean of their absolute element-wise differences with a non-decreasing vector of the same size, which consists of an equal number of copies for each unique value in $\mathcal{D}$.

*JSD and Hellinger distance:*  We discretize the 100k generated data into 100 equal-sized bins between 0 and 1 and compute the frequency of the data in each bin, which provides an empirical probability mass function (PMF). We then compute both the JSD and Hellinger distance between the empirical PMF of the generated data and the empirical (true) PMF of the true (discrete) data.

**Additional Results.**  The observations in Figure 3 are further confirmed by Figure 5, which shows that while Gaussian diffusion has good performance measured by the Wasserstein-1 distance, it is considerably worse than beta diffusion in aligning the supports between the true and generated data distributions. Within beta diffusion, either KLUB or its argument-swapped version leads to good performance measured by both the JSD and Hellinger distance, suggesting their excellent ability to concentrate their generations around the true data supports. However, the proposed KLUB has a much better recovery of the true underlying density in comparison to its argument-swapped version, as confirmed by examining the first subplot in Figure 5 and comparing the second and third subplots in Figure 3.

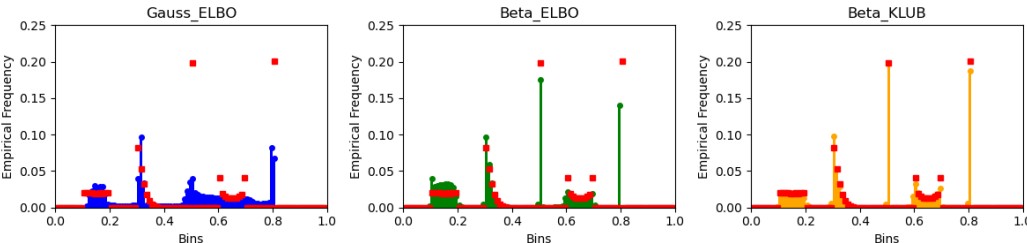

Figure 7: Analogy to Figure 3 for the mixture of continuous range-bounded data and point masses.

Statistical distance between the true and generated data distributions

Figure 8: Analogy to Figure 5 for the mixture of continuous range-bounded data and point masses.

Statistical distance between the true and generated data distributions

Figure 9: Analogy to Figure 8 for comparing "KLUB Conditional," "KLUB Marginal," and "KLUB."

We further conduct an ablation study between KLUB and its two variants: "KLUB Conditional" and "KLUB Marginal," corresponding to $\omega = 1$ and $\omega = 0$, respectively, in the loss given by (18). The results suggest "KLUB Conditional" and "KLUB Marginal" have comparable performance, which is not that surprising considering that in theory, they both share the same optimal solution given by (14). More closely examining the plots suggests that "KLUB Marginal" converges faster than "KLUB Conditional" in aligning the generation with the data supports, but eventually delivers comparable performance, and combining them leads to the "KLUB" that has a good overall performance.

We note we have conducted additional experiments by varying model parameters and adopting different diffusion schedules. Our observation is that while each algorithm could do well by carefully tuning these parameters, "Beta KLUB" is the least sensitive to parameter variations and is consistently better than or comparable to the other methods across various combinations of model parameters. While the toy data is valuable for showcasing the unique properties of beta diffusion, the performance of "Beta KLUB" is not that sensitive to model parameters, and the observed patterns may be disrupted by image-specific settings. Hence, their utility in tuning beta diffusion for image generation, where data dimension and model size/architecture are much larger and more complex, is limited.

## E.1   Mixture of Continuous Range-bounded Distributions and Point Masses

We consider an equal mixture of three range-bounded distributions and two unit point masses, including a Uniform distribution between 0.1 and 0.2, a Beta(1,5) distribution first scaled by 0.1 and then shifted by 0.3, a unit point mass at $x_0 = 0.5$, a Beta(0.5,0.5) distribution first scaled by 0.1 and

then shifted by 0.6, and a unit point mass at $x_0 = 0.8$. A random sample is generated as

$$x_0 = y_k, \ k \sim \text{Discrete}(\{1, 2, 3, 4, 5\}), \ y_1 \sim \text{Unif}(0.1, 0.2), \ y_2 \sim \frac{1}{0.1}\text{Beta}(\frac{y_2 - 0.3}{0.1}; 1, 5),$$

$$y_3 = 0.5, \ y_4 \sim \frac{1}{0.1}\text{Beta}(\frac{y_4 - 0.6}{0.1}; 0.5, 0.5), \ y_5 = 0.9. \tag{29}$$

We follow the same experimental protocol in Section 4.1. The results are shown in Figures 7 and 8, from which we again observe that "Gauss ELBO" does a poor job in aligning its generation with the true data supports and places a proportion of its generation to form smooth transitions between the boundaries of high-density regions; "Beta ELBO" well captures the data supports and density shapes, but has a tendency to overestimate the density of smaller-valued data; whereas "Beta KLUB" very well aligns its generated data with the true data supports and captures the overall density shapes and the sharp transitions between these five range-bounded density regions, largely avoiding placing generated data into zero-density regions.

## F    Preconditioning of Beta Diffusion for Image Generation

It is often a good practice to precondition the input of a neural network. For beta diffusion, most of our computation is operated in the logit space and hence we will consider how to precondition $\text{logit}(z_t)$ before feeding it as the input to the generator $f_\theta$. Specifically, since we have

$$z_t \sim \text{Beta}(\eta\alpha_t x_0, \eta(1 - \alpha_t x_0)),$$

we can draw the logit of $z_t$ as

$$\text{logit}(z_t) = \ln u_t - \ln v_t, \ \ u_t \sim \text{Gamma}(\eta\alpha_t x_0, 1), \ \ v_t \sim \text{Gamma}(\eta(1 - \alpha_t x_0), 1).$$

Assume $x_0 \sim \text{Unif}[x_{\min}, x_{\max}]$, where $x_{\min} = S_{hift}$ and $x_{\max} = S_{cale} + S_{hift}$, we have

$$\mathbb{E}[\text{logit}(z_t)] = \mathbb{E}[\ln u_t] - \mathbb{E}[\ln v_t]$$
$$= \mathbb{E}_{x_0}[\psi(\eta\alpha_t x_0)] - \mathbb{E}_{x_0}[\psi(\eta(1 - \alpha_t x_0))],$$

where

$$\mathbb{E}_{x_0}[\psi(\eta\alpha_t x_0)] = \frac{1}{\eta\alpha_t(x_{\max} - x_{\min})} \int_{\eta\alpha_t x_{\min}}^{\eta\alpha_t x_{\max}} d\ln\Gamma(z)$$
$$= \frac{1}{\eta\alpha_t(x_{\max} - x_{\min})}[\ln\Gamma(\eta\alpha_t x_{\max}) - \ln\Gamma(\eta\alpha_t x_{\min})],$$

$$\mathbb{E}_{x_0}[\psi(\eta(1 - \alpha_t x_0))] = \frac{1}{\eta\alpha_t(x_{\max} - x_{\min})} \int_{\eta(1 - \alpha_t x_{\max})}^{\eta(1 - \alpha_t x_{\min})} d\ln\Gamma(z)$$
$$= \frac{1}{\eta\alpha_t(x_{\max} - x_{\min})}[\ln\Gamma(\eta(1 - \alpha_t x_{\min})) - \ln\Gamma(\eta(1 - \alpha_t x_{\max}))].$$

We further estimate the variance of $\text{logit}(z_t)$ as

$$\text{var}[\text{logit}(z_t)] = \text{var}[\ln u_t] + \text{var}[\ln v_t]$$
$$= \mathbb{E}_{x_0}[\psi^{(1)}(\eta\alpha_t x_0)] + \mathbb{E}_{x_0}[\psi^{(1)}(\eta(1 - \alpha_t x_0))] + \text{var}_{x_0}[\psi(\eta\alpha_t x_0)] + \text{var}_{x_0}[\psi(\eta(1 - \alpha_t x_0))]$$

where

$$\mathbb{E}_{x_0}[\psi^{(1)}(\eta\alpha_t x_0)] = \frac{1}{\eta\alpha_t(x_{\max} - x_{\min})}[\psi(\eta\alpha_t x_{\max}) - \psi(\eta\alpha_t x_{\min})],$$

$$\mathbb{E}_{x_0}[\psi^{(1)}(\eta(1 - \alpha_t x_0))] = \frac{1}{\eta\alpha_t(x_{\max} - x_{\min})}[\psi(\eta(1 - \alpha_t x_{\min})) - \psi(\eta(1 - \alpha_t x_{\max}))],$$

$$\text{var}_{x_0}[\psi(\eta\alpha_t x_0)] = \mathbb{E}_{x_0}[\psi^2(\eta\alpha_t x_0)] - (\mathbb{E}_{x_0}[\psi(\eta\alpha_t x_0)])^2$$
$$\approx \max\left(\frac{1}{100}\sum_{i=0}^{100}\frac{\psi^2\left(\eta\alpha_t\left(x_{\min} + \frac{i}{100}(x_{\max} - x_{\min})\right)\right)}{2^{\mathbf{1}(i=0) + \mathbf{1}(i=100)}} - (\mathbb{E}_{x_0}[\psi(\eta\alpha_t x_0)])^2, \ 0\right),$$

$$\text{var}_{x_0}[\psi(\eta(1 - \alpha_t x_0))] = \mathbb{E}_{x_0}[\psi^2(\eta(1 - \alpha_t x_0))] - (\mathbb{E}_{x_0}[\psi(\eta(1 - \alpha_t x_0))])^2$$

$$\approx \max\left(\frac{1}{100}\sum_{i=0}^{100}\frac{\psi^2\left(\eta\left(1 - \alpha_t\left(x_{\min} + \frac{i}{100}(x_{\max} - x_{\min})\right)\right)\right)}{2^{\mathbf{1}(i=0)+\mathbf{1}(i=100)}} - (\mathbb{E}_{x_0}[\psi(\eta(1 - \alpha_t x_0))])^2, \; 0\right).$$

Now we are ready to precondition $\text{logit}(z_t)$ as

$$g(z_t) = \frac{\text{logit}(z_t) - \mathbb{E}[\text{logit}(z_t)]}{\sqrt{\text{var}[\text{logit}(z_t)]}}.$$

We use $g(z_t)$ as the input to the generator $f_\theta$ and add a skip connection layer to add $g(z_t)$ into the output of the generator. Specifically, following the notation of EDM, we define the network as

$$f_\theta(z_t, t) = c_{\text{skip}}(t)g(z_t) + c_{\text{out}}(t)F_\theta(c_{\text{in}}(t)g(z_t), c_{\text{noise}}(t)),$$

where for simplicity, we set $c_{\text{skip}}(t) = c_{\text{out}}(t) = c_{\text{in}}(t) = 1$ and $c_{\text{noise}}(t) = -\text{logit}(\alpha_t)/8$. A more sophisticated selection of these parameters has the potential to enhance the performance of beta diffusion. However, due to our current limitations in computing resources, we defer this investigation to future studies.

# G   Parameter Settings of Beta Diffusion on CIFAR10

We present the intuition on how we set the model parameters, including $S_{hift}, S_{cale}, \eta, \omega, \pi, c_0$, and $c_1$. Given the limitation of our computation resources, we leave the careful tuning of these model parameters to future work.

We set $\omega = 0.99$ and $\pi = 0.95$ across all experiments conducted on CIFAR10 with beta diffusion.

We pre-process the range-bounded data by linearly scaling and shifting them to lie between $[S_{hift}, S_{hift} + S_{cale}]$, where $0 < S_{hift} < S_{hift} + S_{cale} < 1$. This linear transformation serves two purposes: firstly, it enables us to use beta diffusion to model any range-bounded data, and secondly, it helps avoid numerical challenges in computing the log-beta divergence when its arguments are small. When evaluating the performance, we linearly transform the generated data back to their original scale by reversing the scaling and shifting operations applied during the pre-processing step. For CIFAR10 images, as beta diffusion is diffusing pixels towards $\text{Beta}(0, \eta)$, our intuition is to set both $S_{cale}$ and $S_{hift}$ large enough to differentiate the diffusion trajectories of different pixel values. However, $S_{cale} + S_{hift}$ needs to be smaller than 1, motivating us to choose $S_{cale} = 0.39$ and $S_{hift} = 0.60$.

For the diffusion concentration parameter $\eta$, our intuition is that a larger $\eta$ provides a higher ability to differentiate different pixel values and allows a finer discretization of the reverse diffusion process, but leads to slower training and demands more discretized steps during sampling. We set $\eta = 10000$ and the mini-batch size as $B = 288$ by default, but also perform an ablation study on CIFAR10 with several different values, as shown in Table 3.

We use a sigmoid-based diffusion schedule as

$$\alpha_t = 1/(1 + e^{-c_0 - (c_1 - c_0)t}),$$

where we set $c_0 = 10$ and $c_1 = -13$ by default. We note a similar schedule had been introduced in Kingma et al. [35] and Jabri et al. [29] for Gaussian diffusion.

For the CIFAR-10 dataset[2], we utilize the parameterization of EDM[3] [34] as the code base. We replace the variance preserving (VP) loss and the corresponding network parameterization implemented in EDM, which is the weighted negative ELBO of Gaussian diffusion, with the KLUB loss of beta diffusion that is given by (24). We keep the generator network the same as that of VP-EDM, except that we set $c_{noise} = -\text{logit}(\alpha_t)/8$ and simplify the other parameters as $c_{\text{skip}} = 1$, $c_{\text{out}} = 1$, and $c_{\text{in}} = 1$. The image pixel values between 0 and 255 are divided by 255 and then scaled by $S_{cale}$ and shifted by $S_{hift}$, and hence we have $x_0 \in [S_{hift}, S_{hift} + S_{cale}]$. The two inputs to the generator network $f_\theta(\cdot, \cdot)$, originally designed for VP-EDM and adopted directly for beta diffusion, are $g(\text{logit}(z_t))$ and $\text{logit}(\alpha_t)$, where $g(\cdot)$ is a preconditioning function described in detail in Appendix F. The output of the generator network is first transformed by a sigmoid function and then scaled by $S_{cale}$ and shifted by $S_{hift}$, and hence $\hat{x}_0 = f_\theta(z_t, t) \in [S_{hift}, S_{hift} + S_{cale}]$.

---

[2]`https://www.cs.toronto.edu/~kriz/cifar.html`
[3]`https://github.com/NVlabs/edm`

