# OpenReview forum: "Beta Diffusion"
_NeurIPS.cc/2023/Conference — NeurIPS 2023 poster_

### Official Review · Reviewer_buWN · 2023-07-04

**Soundness:** 3 good
**Presentation:** 2 fair
**Contribution:** 2 fair
**Rating:** 6
**Confidence:** 4

**Summary:**

In this submission, the authors introduce a diffusion model for range-bounded data.
To do so they introduce a noising process that is multiplicative (and not additive) that leads to conditionals that are beta distributed and converges to a beta distribution which is independent of the original datapoint.
They propose to approximate the reverse diffusion by learning the denoised value of $x_0$ given the noised data $z_t$ and the level of noise $\alpha_t$.
They then propose an upper bound on the KL divergence between the forward process and this backward process.
They showcase their model on a range of synthetic 1d datasets.

**Strengths:**

- The proposed approach enables modelling distribution of data which is supported on collection of bounded ranges.
- The beta diffusion process leads to closed form conditionals.
- Since the Kullback-Leibler divergence between Beta distributions can be available in closed-form, the training loss can be computed efficiently.

**Weaknesses:**

- I found this submission not so easy to follow at times. For instance, why is a separate Section 2.4 needed?
- The need to discretised the process to get accurate samples of $q(z_t)$ is a potential drawback of the approach. Is the neural network $f_\theta$ evaluated at each discretisation step during training?
- The experiment are limited to synthetic datasets. It would be interesting to tackle binary label data, or categorical labels via stick-breaking construction as in [Pavel et al. 2023].

Dirichlet diffusion score model for biological sequence generation, Avdeyev, Pavel and Shi, Chenlai and Tan, Yuhao and Dudnyk, Kseniia and Zhou, Jian, 2023.

**Questions:**

- line 123: This seems to hint at the fact that the proposed diffusion models could be written as the discretisation of an SDE which would admits as stationary distribution a Beta distribution. I'd be curious to know whether the Jacobi process used in [Pavel et al. 2023] is that SDE.
- line 154: That's because of the specific choice of parametrisation of $p_\theta(z_s|z_t) = q(z_s|z_t, x_0 = f_\theta(z_t, \alpha_t))$ right? It's quite an interesting remark indeed.
- line 157-158: How samples from marginals $q(z_t)$ are obtained? Does this require discretising the forward process? When is this required? If so what is the added computational cost of this requirement during training?
- line 160: Why starting with $z_1 = 0$? Isn't $z_1 \sim B(0, \eta)$? Does this converge to $\delta_0$ as $\eta \rightarrow 0$? It's worth adding some discussion on this in Section 2.2.
- line 192: What are the consequences of the multiplicative vs additive noise?
-line 227: The difference is solely whether one looks at KL[p|q] or KL[q|p] right? I suppose the theoretical advantage of the latter being that the expectation is taken over the forward process which is easy to sample from. I am not sure to understand the claim that 'they can agree if the first argument of the KL divergence bounded by the KLUB is the PDF of a diffused data distribution'. Any idea why this ELBO loss seems to perform slightly worse? Is it because it's mode seeking?

---

> ### Author Rebuttal · Authors · 2023-08-03
>
> Thank you for your valuable questions. We will now provide detailed clarifications.
>
> >*W1: Why Section 2.4 is needed?*
>
> To implement Algorithm 1 for training beta diffusion, Section 2.4 is not necessary. However, it is needed to demonstrate how discretization functions under beta diffusion. Specifically, Lemma 3 and its proof provide insights into why minimizing Eq. 17 can be seen as an approximate method for minimizing $KL(p_{\theta}(z_{t_{1:T}}) || q(z_{t_{1:T}}))$. Additionally, Eq. 21 illustrates how the trained beta diffusion is used to generate new data with a Markov chain of length $T$.
>
> > *W2: Is there a need to discretize the process to sample from $q(z_t)$? Is $f_{\theta}$ evaluated at each discretisation step during training?*
>
> Beta and Gaussian diffusions share similarities in how their forward and reverse diffusion processes are constructed. As illustrated in Algorithms 1 and 2 in Appendix:
>
> - For the forward process, no discretization is required. We only evaluate $f_\theta$ once at each randomly selected $t$.
>
> - For the reverse process, we perform ancestral sampling to reverse from $t=1$ to $t=0$ in a discrete number of steps, where we need to evaluate $f_\theta$ at each discrete step.
>
> > *W3: The experiment are limited to synthetic data. Comparison with [Pavel et al. 2023].*
>
> We included preliminary experimental results on natural images in the Appendix. While we understand that reviewers were not required to read the Appendix, we appreciate your attention to these results and **Response to All**, where we shared new updates on experiments.
>
> We note that [Pavel Avdeyev et al. 2023] appeared on ArXiv **after** the May 17 submission deadline. We have now carefully reviewed [Pavel Avdeyev et al. 2023] and found distinct differences, kindly refer to **Response to All**.
>
> >*Q1: SDE for beta diffusion?*
>
> We don't believe that the Jacobi process used in [Pavel Avdeyev et al. 2023] corresponds to our beta diffusion for the following reasons:
>
> - First, in beta diffusion, the forward univariate marginal $q(z_t | x_0)$ follows a beta distribution, while Eq. 6 in [Pavel Avdeyev et al. 2023] demonstrates that $q(z_t | x_0)$ in the Jacobi process involves an infinite sum and appears much more complex.
>
> - Second, the endpoint of beta diffusion is $Beta(0, \eta)$, a degenerate beta distribution, rather than $Beta(a, b)$ as seen in  [Pavel Avdeyev et al. 2023].
>
> Nonetheless, there could be potential connections between the proposed beta diffusion and the Jacobi process under specific parameterizations. If you identify clearer connections, we would appreciate your insights and feedback.
>
> >*Q2: Is the analysis in Line 152-154 due to the specific choice shown in Eq. 9?*
>
> The specific choice made in Eq. 9 is justifiable by the analysis in Line 152-154. In essence, optimizing $f_{\theta}(z_t,t)$ aims to make it approach $\mathbb E[x_0|z_t]$. This rationale supports using the particular formulation in Eq. 9.
>
> >*Q3: line 157-158: does forward process require discretization?*
>
> While the forward process is infinitely divisible, as discussed in Line 123-127, we avoid discretization when training. Instead, we can directly transition from $x_0$ to $z_t$ by sampling from a beta distribution, as shown in Eq. 17 and Step 7 in Algorithm 1. Therefore, no discretization is required during training, eliminating any additional discretization cost.
>
> The same as Gaussian diffusion, discretization is only needed during generation.
>
> >*Q4: Why starting from $z_1=0$?*
>
> Following your suggestion, we will add more discussion in Section 2.2:
>
> - By construction, as $t$ approaches 1, $\alpha_t$ approaches 0, leading to $z_t$ approaching $Beta(0,\eta)$, a degenerate beta distribution with a point mass at zero.
>
> - In practice, when the shape of beta distribution becomes small, beta random samples will likely take the smallest positive constant supported by the Float32 precision in PyTorch (i.e., MIN = torch.finfo(torch.float32).tiny=1.1754943508222875e-38). Thus, starting from $z_1$= MIN is common when $\alpha_{t_T}$ is set to be sufficiently close to 0.
>
> >*Q5: Questions on multiplicative vs additive noise,  KL[p||q] vs KL[q||p], KLUB vs ELBO, and why ELBO performs worse for beta diffusion.*
>
> The forward process of beta diffusion, shown in Eqs. 5 and 19, is multiplicative, while Gaussian diffusion's forward process, illustrated in Eq. 31 in Appendix D, is additive. Our toy data experiments favor the multiplicative construction for modeling distributions with sharp transitions between density regions.
>
> The key difference lies in whether we use $KL[p||q]$ or $KL[q||p]$: minimizing the KLUB for $KL( q(z_{t_{1:T}}) || p_{\theta}(z_{t_{1:T}}) )$ corresponds to minimizing -ELBO, while we minimize the KLUB of $KL( p_{\theta}(z_{t_{1:T}}) || q(z_{t_{1:T}}) )$ for beta diffusion. Notably, this choice doesn't impact Gaussian diffusion (Line 220 and Appendix D).
>
> You are correct that $KL( q(z_{t_{1:T}}) || p_{\theta}(z_{t_{1:T}}) )$ is easy to evaluate by sampling from $q(z_{t_{1:T}})$. To evaluate $KL( p_{\theta}(z_{t_{1:T}}) || q(z_{t_{1:T}}) )$ in beta diffusion, we approximate the sampling of $p_{\theta}(z_{t_j} | x_0)$ with $q(z_{t_j} | x_0)$ (Lemma 3 proof).
>
> We note $q(z_{t_{1:T}})$ is the PDF of a diffused data distribution. The statement in Line 232 is trying to say that the KLUB for $KL( q(z_{t_{1:T}}) || p_{\theta}(z_{t_{1:T}}) )$ in beta diffusion becomes equivalent to terms used in -ELBO for optimization.
>
> The inferior performance of -ELBO in beta diffusion arises from the lack of a theoretical guarantee that minimizing it results in $f_{\theta^*}(z_t,t) = \mathbb E[x_0 | z_t]$ (Eq. 13). This relates to the Bregman divergence property. A comparison of Eqs. 12 and 24 highlights distinctions between the proposed KLUB and -ELBO under the log-beta Bregman divergence. Gaussian diffusion is unaffected by using the KLUBs of $KL[p||q]$ or $KL[q||p]$, kindly refer to our response to "*W2*" of Reviewer FX5S.

---

> > ### Author Response · Authors · 2023-08-20
> >
> > Dear Reviewer buWN,
> >
> > We believe that we have effectively addressed all of your previous concerns. We would greatly appreciate any further comments or suggestions you may have, particularly regarding your perspective on the relationship between the proposed beta diffusion and the Jacobi process, which we consider to have developed concurrently. Your feedback is invaluable to us, and we are fully committed to thoughtfully incorporating your insights to enhance our paper.
> >
> > Thanks,
> >
> > Authors of "Beta Diffusion"

---

> > ### Comment · Reviewer_buWN · 2023-08-21
> > **response**
> >
> > Thanks for the detailed response to my comments!
> > This does address my concerns. I'd suggest clarifying / stressing the following points in the manuscript
> > - closed-form forward (no need to discretise)
> > - convergence to $Beta(0, \eta)$ (which is a delta mass)
> > - forward vs reverse KL <=> KLUB vs -ELBO and discussion thereof
> > - connection with [Pavel et al. 2023]
> >
> > I've updated my score accordingly

---

> > > ### Author Response · Authors · 2023-08-21
> > >
> > > We appreciate your taking the time to review our response and for increasing your score. We are delighted to have addressed your concerns and are fully committed to implementing your suggestions, particularly with regard to clarifying and emphasizing these four points.

---

### Official Review · Reviewer_Npra · 2023-07-06

**Soundness:** 3 good
**Presentation:** 3 good
**Contribution:** 3 good
**Rating:** 7
**Confidence:** 4

**Summary:**

This work introduces a new type of diffusion model based on the beta distribution. The end of this is to defined a diffusion model on data that lives inside a specified range, i.e. on an interval [a,b].

This is achieved by first defining a forward noising process conditioned on the initial data sample $x_0$, $q(z_t, z_s | x_0)$ as a bi-variate beta distribution. The marginals of this distribution are also beta distributions, and the conditional $q(z_t | z_s, x_0)$ is also a beta distribution. The distributions are parameterised in such a way that $q(z_0 | x_0) = \delta_{x_0}$ and $q(z_1 | x_0) = \delta_0$. Evolving $t$ from $0 \to 1$ therefore maps between the data distribution and a delta at 0. This process can be seen as a form of continuous diffusion where the noise applied is multiplicative, rather than additive like in Gaussian based diffusions.

They then propose to reverse this distribution by learning the distribution $p(z_s | z_t) = \mathbb{E}_{x_0\sim p_0}[q(z_s | z_t, x_0)] \approx q(z_s | z_t, f_\theta(z_t, \alpha_t))$ and reversing this with small time steps. The learning of this distribution is done by matching $q(z_s | z_t, f_\theta(z_t, \alpha_t)) q(z_t)$ to $ \mathbb{E}[q(z_t, z_s | x_0)] $. This is done via an upper bound on the KL, that can be shown to be minimised when $f_\theta(z_t, \alpha_t) = \mathbb{E}[x_0 | z_t]$. Various other options for KL upper bounds are also proposed to minimise this distance.

The methodology is demonstrated on a couple of simple 1D settings that show the method works well in these cases.

I think this is a very good paper. I would give it a higher score if the preliminary image experiments were finished and added to the main text. I think they would make the paper much stronger - not because they are images per-se but because they show the method works well in high dimension. Other high dimension tasks would be similarly convincing.



**Strengths:**

- The work is cleanly presented and easy to understand.
- There is an extensive exploration of the options available for training these models.
- The work opens up a discussion for other non-additive noise diffusion models, potentially using other infinitly divisible exponential families.
- The experiments are sufficient to convince me that this method does indeed work, and could potentially be practically useful.
- The experimental set up is well explained and looks highly reproducible.

**Weaknesses:**

- I would have liked to see evaluation on some real world data settings to demonstrate the methods utility (in the main paper).
- [1,2] were publicly available about a month before the submission deadline. I appreciate that is close, but I would expect to see at least a related work discussion to these works, which is not present, as they cover the same range-bound data setting as this work. Ideally we would see a comparison to some of the experimental settings in these works.
- I was very surprised to discover image experiments in the appendix after nearly having finished writing this review - these should certainly be mentioned in the main paper (were these completed after the submission but before the supplementary deadline?)

[1] Lou, Aaron, and Stefano Ermon. "Reflected diffusion models." arXiv preprint arXiv:2304.04740 (2023).

[2] Fishman, Nic, et al. "Diffusion Models for Constrained Domains." arXiv preprint arXiv:2304.05364 (2023).

**Questions:**

- Equation (9) is the specified short-time reversal distribution for this diffusion, akin to the Normal distribution in a typical discrete time diffusion model. In that setting the small time reverse diffusion isn't actually a normal, but one can show it converges to a normal with a specific form as the time-step goes to zero, and with a reasonable convergence rate, justifying this choice of variational distribution. Do you have a way of justifying the pick of a beta distribution with the specified form as the small time reverse distribution? For large time steps this obviously cannot be true (e.g. $p(z_0|z_1)$ is certainly not beta distributed unless the data itself is).
- How well does this approach scale with dimension?
- Have the image experiments been finished, and compared properly to other generative models for images?

[1] Lou, Aaron, and Stefano Ermon. "Reflected diffusion models." arXiv preprint arXiv:2304.04740 (2023).

[2] Fishman, Nic, et al. "Diffusion Models for Constrained Domains." arXiv preprint arXiv:2304.05364 (2023).

**Limitations:**

The authors do not really discuss the limitations of their work, and I would like to see this better discussed, particularly with respect to other methods on the same topic. For example unlike score-based SDE models one cannot get a likelihood out of these models.

---

> ### Author Rebuttal · Authors · 2023-08-03
>
> Thank you for providing a comprehensive summary of our paper. We deeply appreciate your positive feedback on the technical details, overall quality, and recognition of the potential for future work. In what follows, we will address any concerns you may have, and we are confident that our response will further emphasize the value and significance of the proposed beta diffusion.
>
> > *W1: No demsontration on real world datasets in the main paper.*
>
> As the camera-ready version permits an extra page, we will relocate the CIFAR10 results from the Appendix to the main paper. Kindly refer to **Response to All** for the latest updates on CIFAR10.
>
> > *W2: Missing discussion on two recent works by [Lou & Ermon, April 2023] and [Fishman et al., April 2023].*
>
> Thank you for bringing these two recent works to our attention. Both of them appeared on ArXiv about a month before the submission deadline. In the camera-ready version, we will include discussions on both papers, as suggested.
>
> Looking ahead, we plan to compare our future extensions of beta diffusion with these works. Specifically, we aim to explore the incorporation of classifier-free guidance (CFG) into logit($z_t$) under beta diffusion and assess its performance in guided image generation, in comparison with [Lou & Ermon, April 2023]. Additionally, we seek to extend the hypercubic constraint, intrinsically satisfied by beta diffusion, to encompass more general manifold constraints, as studied in [Fishman et al., April 2023]. These exciting future endeavors hold great promise.
>
> Upon carefully analyzing both papers, we have noticed several distinctions:
>
> - Both papers attempt to adapt unconstrained Gaussian diffusion to constrained settings, whereas beta diffusion, by design, inherently satisfies the hypercubic constraint.
>
> - While [Lou & Ermon, April 2023] proposes principled methods to handle range-bounded data, and [Fishman et al., April 2023] explores more general manifold constraints (e.g., convex polytope), they are still rooted in the framework of Gaussian diffusion with additive Gaussian noise. By contrast, beta diffusion, with its multiplicative noise, introduces a new foundational hypercubic-constrained diffusion model with potential for further development, such as CFG.
>
> - As of now, there is no evidence of the scalability of [Fishman et al., April 2023] to high dimensions.
>
> For future research, it would be intriguing to investigate how the techniques from [Lou & Ermon, April 2023] and [Fishman et al., April 2023] can be leveraged to enhance beta diffusion for specific applications or broaden its applicability to more general constraints. Such explorations could yield valuable insights and further advancements in the field.
>
> > *W3: Image experiments were not mentioned in the main paper.*
>
> We acknowledge that reviewing supplementary material is at the discretion of the reviewers and sincerely appreciate that you took the time to read our supplementary material before submitting your review.
>
> As indicated in Line 247, more experiments were planned. We had the image generation results on CIFAR10 ready after the main paper deadline. Since these results provided valuable information that directly supported the content of our submission, we believed including them in the Appendix was permissible according to the guidelines outlined in the "NeurIPS 2023 Call for Papers," and therefore, we decided to include them.
>
> Since the initial submission, we have been continuously improving our code by learning from the techniques used by EDM to enhance the training of Gaussian diffusion. Some of these techniques have proven beneficial for beta diffusion as well, resulting in a clear improvement in its FID on CIFAR10. Our latest FID achieved is 3.56. For further details, please refer to **Response to All**.
>
> > *Q1: Justification for Eq.9 under large-step time reversal.*
>
> We would like to address your question from two perspectives:
>
> - First, the justification of Eq. 9 can be found in Line 150-155, demonstrating that it provides an approximation to the true time reversal distribution, given by $q(z_s | z_t) = \mathbb E_{x_0\sim q(x_0|z_t)}[q(z_s|z_t,x_0)]$. The optimization of $\theta$ results in Eq. 9 becoming equivalent to $p_{\theta^*}(z_s|z_t) = q(z_s|z_t,\mathbb E[x_0|z_t])$.
>
> - Second, while Gaussian diffusion uses a small-step time reversal analysis for its justification, we have noted that the specific parameterization utilized in reverse Gaussian diffusion can also be justified by letting $p(z_s|z_t) = q(z_s| z_t, \hat x_0 = f(z_t,t) )$, as shown in Eq. 9 of [Kingma et al. 2021, Variational Diffusion Models]. This reference, which was already cited in our paper, will be further highlighted to support the use of Eq. 9 in reverse beta diffusion.
>
> > *Q2: How well does this approach scale with dimension?*
>
> Our image experiments indicate that beta diffusion scales with dimension similar to Gaussian diffusion. While we have already tested beta diffusion on CIFAR10, we are also planning to conduct experiments with higher-resolution images. As detailed in Line 541-543 in Appendix F, using a machine equipped with four Nvidia RTX A5000 GPUs, both beta diffusion and Gaussian diffusion (VP-EDM) take approximately 1.46 seconds to process 1000 images of size 32×32×3 during training.
>
> > *Q3: Have the image experiments been finished, and compared properly to other generative models for images?*
>
> We have continually worked on improving our training code by incorporating specific techniques used by EDM to enhance the training of Gaussian diffusion (DDPM, VP-EDM). However, we have yet to thoroughly explore methods to further enhance the current beta reverse diffusion sampler, which we aim to investigate in future research. For our most recent FID results on CIFAR10 and a comparison with representative non-Gaussian and Gaussian-like diffusion models, kindly refer to **Response to All**.

---

> > ### Comment · Reviewer_Npra · 2023-08-22
> >
> > I want to thank the authors for their replies.
> >
> > I am happy with the points made, and will keep my score as accept as-is.

---

> > > ### Author Response · Authors · 2023-08-22
> > >
> > > We are pleased to hear that our responses meet your expectations and appreciate your continued support!

---

### Official Review · Reviewer_FX5S · 2023-07-08

**Soundness:** 3 good
**Presentation:** 3 good
**Contribution:** 3 good
**Rating:** 6
**Confidence:** 5

**Summary:**

This paper addresses a prevalent assumption found in deep generative diffusion models, which is the Gaussian assumption in both the forward and reverse processes. In this study, the authors explore the utilization of the beta distribution in these processes and establish several key properties. Firstly, they analytically compute the marginal distribution within the beta distribution framework. Additionally, they develop an analytical computation for the conditional distribution, given both the data point and a noisy sample, in the form of a scaled and shifted beta distribution. To optimize the beta diffusion process, the authors propose a combination of two distinct KL upper bounds, which is better than using the standard ELBO. The experimental results on a synthetic setup demonstrate the ability of the beta distribution to effectively model range-bounded data distributed across disjoint regions. Furthermore, they show the efficacy of the proposed objective functions. The supplementary material also includes preliminary experiments conducted on image generation tasks.

**Strengths:**

[Uniqueness of the methodology]

Unlike previous works on diffusion-based generative models that heavily rely on the Gaussian assumption for both the forward and reverse processes, this work stands out as the first attempt to introduce Beta distributions into these processes. This unique approach sets it apart from existing literature. Furthermore, the authors have devised an objective function specifically tailored to optimize the beta diffusion process, further distinguishing their methodology from previous approaches.

[Presentation]

The manuscript is overall well-written and exhibits clarity in its presentation. Particularly, the introduction and related works sections provide concise and comprehensive summaries of prior research, effectively highlighting the distinction between the Gaussian diffusion process and the proposed Beta distribution process. To further enhance the motivation behind this work, it would be beneficial to include a practical scenario that illustrates the necessity of constructing the Beta diffusion process. This addition would greatly strengthen the motivation behind the study.

**Weaknesses:**

[Absence of real-world experiments]

The main paper primarily presents experiments conducted on a synthetic dataset. While strong theoretical analysis can support the use of synthetic experiments, including more real-world experiments would further reinforce the proposed methods. It would be beneficial to provide practical scenarios that demonstrate why the beta diffusion process should be considered over the well-established Gaussian diffusion. In the supplementary material, preliminary experiments on image generation tasks are included, indicating potential in that direction. However, it is not entirely clear why the image generation task specifically requires a beta diffusion process. One could argue that since images are typically bounded within the range [0, 1], the beta distribution is more suitable. Theoretical reasoning supports this argument. However, in practice, simply clipping the values during the reverse steps is often sufficient to remove outliers during sampling.

[KLUB]

I remain unconvinced about the superiority of KLUB over the negative ELBO in optimizing the beta diffusion process. Could you provide further elaboration in the rebuttal? Additionally, I am curious if this argument can be generalized to the standard Gaussian diffusion process as well.


**Questions:**

[About the formulation]

#1. One area where the real benefits of Beta diffusion could be highlighted is in guided sampling. In classifier-free guidance, as well as classifier-guided sampling, the presence of outliers can lead to low-quality generated samples, particularly in cases with high guidance scale. This typically leads to the generated samples of high-contrast. This issue is commonly observed in many open-source diffusion models such as SD, Karlo, and IF. The Imagen paper by Google addresses this problem by employing a dynamic threshold heuristic. However, I believe that Beta diffusion could serve as a fundamental solution to mitigate this drawback in guided sampling. Due to the design of the diffusion process, the data range is naturally constrained, making Beta diffusion well-suited to address this challenge.

#2. Is it possible to extend this framework to the exponential family? If so, it would allow for a unified perspective encompassing Gaussian, Categorical, and Beta diffusion processes. This extension could contribute to a more comprehensive understanding of diffusion models.

#3. In image generation tasks, various approaches, including Stable Diffusion, train diffusion models on the latent space learned by an autoencoder. I believe that Beta diffusion could be readily applied to the latent space with relative ease, but I would be interested to hear the authors' opinion on this matter. It is worth noting that in the latent dimension, the range is no longer limited to [0, 1], but it can be regulated by applying appropriate regularization techniques to the stage 1 model, such as the autoencoder in LDM.

[About the experiments]

#1. Have you considered comparing the simplified denoising loss, which discards the weights in the ELBO, to Beta KLUB? The DDPM paper has shown that this simplified loss outperforms the reweighted loss in image generation tasks. I'm curious if you observed similar results in your experiments.

#2. Did you clip the values during the sampling steps in Gaussian diffusion to the pre-defined range ([0,1] or [-1, +1])? In practice, clipping the values during the reverse steps can improve the quality of the samples, especially when dealing with high CFG scale.

#3. It would be helpful to provide more practical scenarios where the Beta diffusion process is more suitable compared to the vanilla Gaussian diffusion. While the supplementary material includes a preliminary experiment on image generation tasks, in my opinion, the Gaussian diffusion process already works quite well in the image domain. Therefore, showcasing more compelling applications of the Beta diffusion process would significantly strengthen the motivation behind your work.


**Limitations:**

The paper does not explicitly outline its limitations. My suggestions to improve this work are described in the sections above. As this work is primarily theoretical in nature, it is unlikely to have any significant negative societal impacts.

---

> ### Author Rebuttal · Authors · 2023-08-04
>
> We appreciate your excellent summary of the key contributions of our paper and your recognition of the uniqueness of the proposed methodologies. We are delighted that you found our paper to be well-written and presented with clarity. Below please see our point-by-point response.
>
> > *W1: Absense of real-world experiments in the main paper.*
>
> In **Response to All**, we demonstrate that beta diffusion now surpasses many existing non-Gaussian diffusion models in generating CIFAR10 images, and approaches Gaussian ones.
>
> Regarding the heuristic of "clipping" in reverse Gaussian diffusion, we acknowledge its widespread use, especially under classifier-free guidance (CFG). However, the reference [Lou & Ermon, April 2023], pointed out by Reviewer Npra, aims to eliminate the need for this heuristic in Gaussian diffusion. Beta diffusion offers a principled alternative to avoid this heuristic by performing CFG in the logit space that our Unet operates on, which we plan to investigate as part of our future work.
>
> We also recognize the value of evaluating beta diffusion in various tasks where its multiplicative construction and hypercubic-constrained nature could provide unique advantages. We eagerly anticipate collaborating with domain experts in different applications, such as modeling high-dimensional biological data, which often exhibit range-bounded, sparse, skewed, and heavy-tailed characteristics.
>
> > *W2: Is KLUB superior over -ELBO?*
>
> Yes, please see Table 2 in **Response to All**. We acknowledge the need for clarifying the four different types of KLUBs:
>
> - KLUB-conditional: A KLUB of $KL[p_{\theta}(z_s|z_t)q(z_t)||q(z_s,z_t)]$, as shown in Eq.10-11.
> - KLUB-marginal: A KLUB of $KL[p_{\theta}(z’_t)||q(z’_t)]$, as shown in Eq.14-15
> - KLUB-conditional-AS (AS stands for augment-swapped): A KLUB of $KL[q(z_s,z_t)||p_{\theta}(z_s|z_t)q(z_t)]$, which is the same as Eq.11 except for swapping the two arguments inside the KL.
> - KLUB-marginal-AS: A KLUB of $KL[q(z_t')||p_{\theta}(z_t')]$, which is the same as Eq.15 except for swapping the two arguments inside the KL.
>
> It is important to note that optimizing with KLUB-conditional-AS is equivalent to optimizing with -ELBO. All four KLUB types can be expressed using the log-beta Bregman divergence. However, only the first two KLUBs guarantee the optimal solution as shown in Eq.13. Using $w$KLUB-conditional-AS+$(1-w)$ KLUB-marginal-AS or -ELBO does not provide such a guarantee. This explains why the proposed KLUB (i.e., $w$ KLUB-conditional + $(1-w)$ KLUB-marginal) is found to be better than KLUB-AS or -ELBO for optimizing beta diffusion.
>
> For Gaussian diffusion, we find interesting theoretical justifications for the SNR weighted -ELBO, which is often considered a heuristic but crucial modification of -ELBO:
>
> - As shown in Appendices D.1-D.2, in Gaussian diffusion, KLUB-conditional = KLUB-conditional-AS = -ELBO.
> - As shown in Appendix D.3, KLUB-marginal = KLUB-marginal-AS = SNR weighted -ELBO.
>
> This result arises from the symmetric nature of the squared Euclidean distance, which is the Bregman divergence under $KL[p||q]=KL[q||p]$, where $p$ and $q$ are univariate Gaussians with equal variance. The findings shed light on the justified use of SNR weighted -ELBO in Gaussian diffusion.
>
>
> > *F1: Potenial of beta diffusion for guided sampling.*
>
> Thank you for highlighting the potential benefit of beta diffusion. In our future work, we plan to explore beta diffusion with label/text guidance, with/without CFG, and compare its performance to the principled solution for guided Gaussian diffusion proposed by [Lou & Ermon, April 2023].
>
> >*F2: Is it possible to extend this framework to the exponential family?*
>
> Yes, we are actively working on the extension to infinitely divisible exponential family distributions. Notably, we have already addressed the technical aspects for both the exponential and Gamma distributions, two additional members of the exponential family, and are in the process of conducting experiments with them.
>
> > *F3: Latent beta diffusion?*
>
> Thanks for the suggestion! We'll explore using sigmoid/tanh activation in the encoder's last layer to provide a bounded latent space for beta diffusion to work on. This will be valuable for scaling beta diffusion to high-resolution images in our future work.
>
> > *E1: Simplified loss for beta KLUB?*
>
> Eq. 1 demonstrates that dropping the $\alpha_t$-related weight coefficients in beta ELBO/KLUB is not straightforward. For Gaussian diffusion, we default to using the SNR weighted -ELBO (i.e., simplified -ELBO). As explained in our response to "*W2*", in Gaussian diffusion, KLUB-conditional = -ELBO and KLUB-marginal = simplified -ELBO. Therefore, we can interpret the comparison between KLUB-conditional and KLUB-marginal as analogous to comparing the original loss with the simplified loss.
>
> Our toy experiments indicate that KLUB-marginal often converges faster, and combining both KLUB-conditional and KLUB-marginal yields the best overall performance.
>
> > *E2: Clipping or not?*
>
> We did not apply clipping in the toy data experiments. When quoting the unconditional CIFAR10 image generation results of Gaussian diffusion from their original papers, we considered the absence of CFG. Additionally, we assumed that if clipping was important, the original models had already applied it.
>
> >*E3: Where beta diffusion are more suitable than Gaussian diffusion?*
>
> Gaussian diffusion was initially introduced in 2015 but only gained prominence with DDPM due to its optimized combination of architecture, loss modification, and noise scheduling. We believe that beta diffusion holds promise for image generation, particularly with proper customization and tuning of its network architecture and hyperparameters, and can remove the need for heuristic clipping under CFG. In scenarios where Gaussian-like distributions are unsuitable for modeling, such as many biological datasets, we anticipate that beta diffusion could prove highly advantageous.

---

> > ### Comment · Reviewer_FX5S · 2023-08-16
> > **Maintain original Score; Recommend acceptance**
> >
> > I am grateful for the authors' comprehensive response to my comments. They have effectively addressed the main concerns I raised in the initial reviews. Furthermore, it is encouraging to learn that the authors are actively exploring potential extensions of the beta diffusion framework. This includes its application to the exponential family as well as its integration with latent diffusion models. With these positive developments in mind, I am confident in maintaining my original score and recommend the acceptance of this work.

---

> > > ### Author Response · Authors · 2023-08-16
> > >
> > > Your thoughtful assessment is greatly appreciated. We are pleased that our response has effectively addressed your main concerns. Your feedback will be thoughtfully incorporated to enhance the quality of our paper. We are particularly encouraged by your positive reception of our exploration into potential extensions of the beta diffusion framework. The extension to the exponential family and the integration with latent diffusion models hold exciting possibilities, and we are fully committed to advancing these aspects further. Thank you for your valuable input.

---

### Official Review · Reviewer_n1uL · 2023-07-26

**Soundness:** 3 good
**Presentation:** 2 fair
**Contribution:** 3 good
**Rating:** 6
**Confidence:** 3

**Summary:**

The presented work proposes to design the forward and reverse process of diffusion models based on beta distributions (as opposed to Gaussian noise). It also showcases a new loss term based on KL-divergence upper bounds (KLUB) that can aid performance over the conventional ELBO-based diffusion model objective when used to optimize the proposed beta diffusion model. The paper shows that the proposed beta diffusion model can better model/generate synthetic distributions, especially point masses, compared to Gaussian diffusion, but lags behind it when used for CIFAR-10 image generation.

**Strengths:**

- The paper is well-written
- Alternatives to Gaussian diffusion processes are an interesting and relevant direction
- The synthetic experiments give a good intuition on the advantages of beta diffusion over Gaussian diffusion (mostly in terms of modeling point masses)


**Weaknesses:**

- Comparison and discussion with related work is lacking. As a result, the novelty of the presented work is hard to assess. While not having a related work section may be okay in some cases, discussing relevant related work thoroughly is very important. In particular, the presented paper completely misses to discuss or even cite any closely related papers on alternative diffusion processes of which at least [1, 2, 3, 4] come to my mind.
- In addition, a quick literature search on my end popped out [5] (ICML 23), which to me seems like  extremely related work. [5] also proposes to use Beta distributions to design a novel diffusion process (and even go beyond that by extending it to a Dirichlet distribution-based diffusion model). While to be fair [5] can be deemed concurrent work to the one presented (and was uploaded to Arxiv only in May), I think that it is absolutely key to discuss this paper, outline how the presented work is different (or not) from it, and potentially even benchmark against it (in [5] they show that they outperform D3PM, which is a baseline in this paper too).
- There are multiple passages in the draft (e.g. line 78, 238, 346) that make it sound like KLUB-based objectives can be used for diffusion models in general, but there is no experiment backing this claim and it directly contradicts the authors themselves (lines 219-225). This seems like overclaiming to me.
- Experiments are not entirely convincing. While I see the value of the presented synthetic experiments, I believe that a more comprehensive evaluation is required. Some concerns that I have are:
    1) Both synthetic experiments seem to show to me that beta diffusion can model point masses better than Gaussian diffusion. This is great, but makes me question the value of having two synthetic toy datasets in the main paper that lead to the same conclusion, especially compared to including more challenging datasets (Cifar10 experiments are only shown in the appendix and not mentioned in the main text).
   2) The lack of other convincing experiments makes it unclear how beta diffusion would perform on non-toy datasets (basically anything except for point masses and beta distributions), and it is totally nebulous in which cases, if any, beta diffusion could be a strongly performing alternative to other diffusion models (on cifar10, beta diffusion FID scores are worse than the ones from Gaussian diffusion or [2, 3]). The paper would strongly benefit from a more realistic example that showcases where beta diffusion may have an edge over other diffusion models on any relevant metric (does not need to be state-of-the-art performance).
   3) The authors can do a better job of describing how and why anyone should use a beta diffusion model. For example, the hyperparameter values for the synthetic experiments are quite different from the ones used for Cifar10 (some beta diffusion-specific ones include the diffusion concentration parameter and how to bound the data) , and there is no discussion on how to choose them in practice (or even how the authors arrived at their particular choice of values).
   4) The synthetic datasets would seem to be a great opportunity for a detailed exploration and/or ablation of the design space for beta diffusion. However, there are only few ablative studies presented, and for the ones presented it is unclear how statistically significant they are, since no error bars are shown (esp. Figures 2,3,5 should have them) . One ablative study that would be important to include is ablating \beta_max from the appendix – To my understanding, when \beta_max is chosen to be too large, the end distribution (of z_1) will be just a black image, in which case there would be no stochasticity at all coming from the initial conditions during the reverse sampling process, which implies that a trade-off exists that is important to understand. Additionally, it would be valuable to ablate using ELBO vs KLUBs as beta diffusion objective on a non-toy dataset (e.g. Cifar10), as well as ablating NFE (i.e. #iterations during inference) for any trained (fixed) beta diffusion model.

[1] https://arxiv.org/abs/2208.09392

[2] https://research.google/pubs/pub52492/  (TMLR 23)

[3] https://openreview.net/forum?id=OjDkC57x5sz (ICLR 23)

[4] https://openreview.net/forum?id=4PJUBT9f2Ol  (ICLR 23)

[5] https://arxiv.org/abs/2305.10699 (ICML 23)


**Questions:**

- What does \beta_max in the appendix correspond to in the main text?
- How many generated samples are used to compute the Cifar10 FID?
- More direct comparison to Gaussian diffusion on CIFAR 10 would be valuable (even if on a subset of Cifar10)
- In terms of computational efficiency, are there any differences between beta and Gaussian diffusion?
- The authors mention that numerical instabilities can arise in fp32 for beta diffusion, and it would be valuable to quantify this (e.g. show plot that shows which hyperparameter ranges may cause that).

*************************************************
After the rebuttal, I would like to raise my score to 6 and recommend acceptance. I'm looking forward to seeing the new experiments, related work discussion & benchmarking, and ablations in the revised paper.

**Limitations:**

The paper would benefit from a more thorough discussion of the limitations of the presented work. As far as I can tell, the only limitation mentioned is that no hyperparameter tuning was performed, which is totally okay, but is not a limitation of the method itself (which would be much more valuable and interesting to read about).

---

> ### Author Rebuttal · Authors · 2023-08-04
>
> The reviewer found our paper to be well-written and the exploration of beta diffusion as an interesting and relevant direction, but was very concerned about the lack of discussion and comparison with related works. We aim to decisively alleviate these concerns with our response and revision.
>
> > *W1: Related work.*
>
> In Introduction, we carefully reviewed key references on Gaussian diffusion, effectively highlighting how beta diffusion is distinct (kindly refer to Reviewer FX5S).
>
> We appreciate your mention of [1-4] as related works on alternative diffusion processes. In **Response to All**, we have included FID comparison to [1-4] on CIFAR10, showing that beta diffusion now outperforms all except [4]. Below, we further discuss how beta diffusion is distinct from [1-4] to emphasize its novelty:
>
> - [1]: Cold Diffusion builds models around arbitrary transformations (e.g., blurring) instead of Gaussian corruption. [2]: Soft Diffusion uses linear corruption processes (e.g., Gaussian blur and masking).  [4]: Inverse Heat Dispersion explicitly reverses the heat equation by exploring inductive biases in Gaussian diffusion-like models.
>
> - [3]: Extending [4], Blurring Diffusion adds blurring (or heat dispersion) into Gaussian diffusion.
>
> - These alternative diffusion processes in [1-4], Blurring Diffusion in particular, still resemble Gaussian diffusion in not only how the squared Euclidean distance is used to define the training loss but also their use of Gaussian-based reverse diffusion for generation.
>
> - Beta diffusion, on the other hand, is distinct from [1-4] in forward diffusion, training loss, and reverse diffusion. Its loss can be expressed as a log-beta Bregman divergence, which theoretically guarantees that $f_{\theta^*}(z_t,t)=\mathbb E[x_0|z_t]$.
>
> > *W2: [5] is extremely related but not cited.*
>
> Due to [5 (Pavel Avdeyev et al., ICML 2023)] appearing on ArXiv after the May 17 submission deadline, we were unaware of its existence.
>
> Therefore, we kindly request you to reconsider its relevance in evaluating beta diffusion. Furthermore, in our reading of [5], we noted significant differences, as elaborated in **Response to All**.
>
> > *W3: Can KLUBs be used for diffusion models in general?*
>
> Indeed, KLUB-based objectives are applicable to diffusion models in general, including both beta and Gaussian diffusions. We believe this aspect was carefully addressed without any self-contradiction (Line 219-225) or overclaiming. For further clarification, please refer to Appendix D, showing how KLUB-based objectives relate to (weighted) -ELBO for Gaussian diffusion, and also our response to “*W2*” from Reviewer FX5S.
>
> > *W4-1:*
>
> The 2nd toy data differs from the 1st in that it also includes range-bounded continuous variables. We will utilize the extra page available in Camera Ready to include the updated CIFAR10 results in the main paper.
>
> >*W4-2:*
>
> As in **Response to All**, beta diffusion now outperforms non-Gaussian diffusion models, including [1,2,4], on CIFAR10, and is competitive against Gaussian diffusion with or without adding blurring [3].
>
> >*W4-3:*
>
> We will release the code along with detailed instructions to enable domain experts to explore beta diffusion on their own data and applications.
>
> - For toy data, we set the concentration parameter as $\eta=1000$, while for images, to distinguish between 256 pixel values, we choose a larger value of $\eta=10000$. A larger $\eta$ might result in slower training and the need for a finer discretization of reverse diffusion, with FIDs on CIFAR10 differing by approximately 1 between $\eta=10000$ and $\eta=1000$.
>
> - As beta diffusion is diffusing pixels $x_0$ towards $Beta(0, \eta)$, our intuition suggests setting both Scale and Shift large enough to differentiate the diffusion trajectories of different pixel values. With this in mind, [Shift=0.5, Scale=0.4] provides a good compromise.
>
> - It is common practice to choose data and network-architecture specific parameters for diffusion models. While ideally, we would perform a grid search of these parameters, we haven't done so for beta diffusion yet due to limited computing resources.
>
> >*W4-4:*
>
> We appreciate your suggestion regarding exploring/ablating the design space on toy datasets. While they are valuable for code debugging and showcasing the unique properties of beta diffusion, their performance is not that sensitive to model parameters, and the observed patterns may be disrupted by image specific settings. Hence, their utility in tuning beta diffusion for image generation, where data dimension and model size/architecture are much larger and more complex, is limited.
>
> - For toy data, we did not include error bars as the performance gaps between KLUB and -ELBO were large. We are glad to add them into Camera Ready.
>
> - For both Gaussian and beta diffusions, selecting an appropriate $\beta_{\max}$ is desirable. In Gaussian diffusion, an overly large $\beta_{\max}$ can lead to many wasted diffusion steps at $x_t\sim N(0,1)$. Similarly, in beta diffusion that starts its reverse from $x_1\sim Beta(0,\eta)$, a point mass at 0, if $\beta_{\max}$ is too large, many training and reverse steps for $t$ close to 1 will be wasted at $x_t=$ MIN = torch.finfo(torch.float32).tiny= 1.1754943508222875e-38, the minimum threshold in sampling Beta in PyTorch Float32 precision.
>
> - **Response to All** now includes -ELBO vs KLUB on CIFAR10 and FIDs under different NFEs.
>
> > *Q 1-5:*
>
> - The main paper employs the sigmoid schedule and, therefore, does not include $\beta_{\max}$, which is a parameter of the beta linear schedule used by DDPM (see Algorithm 1).
>
> - As per convention, we used 50,000 generated samples to compute FID.
>
> - Please see **Response to All**.
>
> - No clear difference. See Line 542 in Appendix F.
>
> - Numerical instabilities may arise when Shift approaches zero. This is not surprising, as KL[Beta($a,b$)||Beta($c,d$)] exhibits numerical challenges when either $a$ or $c$ is close to zero.

---

> > ### Comment · Reviewer_n1uL · 2023-08-17
> > **Raise score and recommend acceptance**
> >
> > Thank you for the convincing rebuttal, especially regarding the comparison to [1-4] in terms of both methodological and performance differences. I will raise my score and recommend acceptance as a result. Optimally, it would have been beneficial if the related work discussion and the Cifar 10 experiments had been ready for the main paper submission deadline. Please include those in the revised paper, incl. your verbal comparison with refs [1-5] (I think that discussing [5] in the paper would be great, even if it was published after your first submission for Neurips).
> >
> > > Indeed, KLUB-based objectives are applicable to diffusion models in general, including both beta and Gaussian diffusions. We believe this aspect was carefully addressed without any self-contradiction (Line 219-225) or overclaiming. For further clarification, please refer to Appendix D, showing how KLUB-based objectives relate to (weighted) -ELBO for Gaussian diffusion, and also our response to “W2” from Reviewer FX5S.
> >
> > I understand that KLUB can be used as an objective for Gaussian diffusion too. But, as you note in multiple places (e.g. appendix D), this seems to lead to the same existing objective (-ELBO). What I am complaining about in the referenced lines (e.g 78, 238, 346) is that they make it sound to me as though KLUB is a new way of optimizing diffusion models *in general*, even though practically it only leads to a new (and more effective, indeed) way of optimizing beta diffusion. As such, in my opinion it would be better to only claim having shown that KLUBs are an effective objective for *beta* diffusion models.
> >
> > > For toy data, we set the concentration parameter as, while for images, to distinguish between 256 pixel values, we choose a larger value (...)
> >
> > Please include this ablation of the concentration parameter on Cifar 10 to the revised paper.

---

> > > ### Author Response · Authors · 2023-08-17
> > >
> > > We deeply appreciate your careful consideration of our rebuttal, resulting in a higher score and a recommendation for acceptance. Following your guidance, we will ensure including the CIFAR10 results into the main paper, adding careful discussions and comparisons with references [1-4], and highlighting the connections and distinctions with [5].
> > >
> > > Your detailed clarification regarding your concerns about our portrayal of KLUBs is genuinely valued. One of the driving factors behind our enthusiasm for KLUBs was their applicability to Gaussian diffusion, allowing us not only to derive the original -ELBO but also the SNR-weighted -ELBO. This enhancement, often regarded as a heuristic but important modification of the original -ELBO, can now be theoretically grounded with the use of KLUBs. Additionally, we have identified another case (a gamma distribution based diffusion model) where KLUBs, based on a Bregman divergence-centered rationale, would outperform -ELBOs. Admittedly, as our current paper is focused on beta diffusion, we concur with your recommendation to restrict our claims regarding KLUBs to the domain of beta diffusion models.
> > >
> > > We will follow your suggestion to include an ablation study on the concentration parameter using CIFAR10 in the revised version of our paper. Your insights are very helpful in guiding us towards areas of improvement, and we will integrate them to carefully revise our paper. Once again, we appreciate your valuable input.

---

### Author Rebuttal · Authors · 2023-08-04

# Response to All

We express our gratitude to all four reviewers for their valuable insights and suggestions. Their comments have been instrumental in identifying relevant recent works on alternative diffusion processes, refining the paper's positioning within the literature, elucidating the properties of KLUBs in relation to (weighted) ELBOs, and enhancing comparison studies. We will diligently incorporate these enhancements into the final version of the paper.

The feedback from the reviewers reflects their overall satisfaction with the technical aspects of the paper. Nevertheless, the absence of certain related works concerning alternative diffusion models has prompted legitimate concerns. This absence has hindered a comprehensive evaluation of the distinct contributions that our proposed beta diffusion model offers to the rapidly evolving landscape of diffusion models. Although we elucidated the differences between beta diffusion and Gaussian diffusion in our paper, we acknowledge that a more comprehensive discussion and comparison with alternative diffusion models would enhance the contextualization of our work.

To address these concerns effectively, we have taken care to discuss and compare to related works on alternative diffusion processes and provide thorough responses to each reviewer. Furthermore, we extend our explanations below to provide additional clarity, helping the reviewers in their assessment of the beta diffusion model's novelty, significance, and potential for generalization. We also emphasize the adaptability of the optimization techniques associated with the beta diffusion model, suggesting their potential applicability to a broader spectrum of exponential family distributions.

1. Following submission, we have refined our code by incorporating design strategies from EDM that enhance the training of Gaussian diffusion. Specifically, we pre-condition logit($z_t$) with its mean and standard deviation before feeding it into the Unet $f_{\theta}$ for CIFAR10 image experiments. This refinement has notably improved the FID score. Our initial reported FID of 6.17 in the Appendix has been lowered to **3.56** due to these improvements. Additionally, we are keen on exploring EDM's techniques to enhance our reverse beta diffusion sampler in future research endeavors.

2. Our analysis in Table 1 within the attached PDF encompasses a broad spectrum of diffusion models, both Gaussian and non-Gaussian. We highlight that beta diffusion now outperforms all non-Gaussian diffusion models on CIFAR10, including Cold Diffusion and Inverse Heat Dispersion, as well as categorical and count-based diffusion models. In comparison to Gaussian diffusion and Gaussian+blurring diffusion, beta diffusion surpasses Soft Diffusion. While it may fall short of DDPM, VP-EDM, and Blurring Diffusion, it remains a competitive alternative that uses non-Gaussian based diffusion.

3. Table 2 within the attached PDF has been introduced to compare KLUB and negative ELBO-optimized beta diffusion across various NFEs. We also include a Figure to visually compare generated images under KLUB and ELBO. The findings presented in Table 2 and the Figure provide further validation of KLUB's efficacy in optimizing beta diffusion.

4. We acknowledge the mention of [Pavel Avdeyev et al., 2023] by two reviewers. As this paper emerged on ArXiv post the May 17 submission deadline, we deem it concurrent work. We kindly request reviewers to reconsider its relevance in evaluating our paper. Furthermore, our analysis of the paper reveals key distinctions:

    - [Pavel Avdeyev et al., 2023] employs the Jacobi diffusion process for discrete data diffusion models. In contrast to Gaussian diffusion's SDE definition, the Jacobi diffusion process follows $dx = \frac{s}{2}[a(1-x)-bx]dt+\sqrt{sx(1-x)}dw$, with $x\in[0,1]$ and $s,a,b>0$. Its stationary distribution is a univariate beta distribution $\mbox{Beta}(a,b)$.

   - While beta diffusion and the Jacobi process are connected by the beta distribution, they differ in various aspects:

       - Beta diffusion concludes its forward process at $\mbox{Beta}(0,\eta)$, a unit point mass at 0, not a $\mbox{Beta}(a,b)$ random variable.

        - The marginal distribution of beta diffusion at time $t$ is denoted as $q(z_t| x_0)\sim \mbox{Beta}(\eta\alpha_t x_0,\eta(1-\alpha_t x_0))$, distinct from the infinite sum in the Jacobi diffusion process. The potential links between beta diffusion and the Jacobi process, under specific parameterizations, warrant further exploration.

    - The Jacobi diffusion process and its stick-breaking-based multivariate extension primarily address discrete data (binary and categorical) in relatively low dimensions. Their adaptability to range-bounded continuous data and scalability to high dimensions remain unestablished. In contrast, beta diffusion effectively models range-bounded continuous data and scales comparably to Gaussian diffusion in higher dimensions.

5.  Two of the reviewers highlighted the promising prospect of integrating classifier-free guidance (CFG) into beta diffusion, without resorting to heuristic clipping. We share the reviewers' enthusiasm regarding this potential advancement, and we believe that beta diffusion is poised to seamlessly integrate CFG by applying it to the logit space that our Unet operates on. We extend our gratitude to the reviewers for pointing out this future research direction, which we are eager to explore and further develop.

---

### Decision · Program_Chairs · 2023-09-21

**Decision:**

Accept (poster)

**Comment:**

The authors develop a method for diffusions that model data in bounded support. Their approach is based on multiplicative transitions derived from the beta distribution. They develop an upper bound on the KL divergence to train this model. The approach is fresh and well-described. All of the reviewers are in agreement that the paper should be accepted.  In the final version, the authors should update their related work especially to include the concurrent piece of work (Dirichlet diffusion score model for biological sequence generation, Avdeyev et al., ICML 2023) that was discussed.